# Interaction-Grounded Learning
# with Action-Inclusive Feedback

**Tengyang Xie**[*]
UIUC
tx10@illinois.edu

**Akanksha Saran**[*]
Microsoft Research, NYC
akanksha.saran@microsoft.com

**Dylan J. Foster**
Microsoft Research, New England
dylanfoster@microsoft.com

**Lekan Molu**
Microsoft Research, NYC
lekanmolu@microsoft.com

**Ida Momennejad**
Microsoft Research, NYC
idamo@microsoft.com

**Nan Jiang**
UIUC
nanjiang@illinois.edu

**Paul Mineiro**
Microsoft Research, NYC
pmineiro@microsoft.com

**John Langford**
Microsoft Research, NYC
jcl@microsoft.com

## Abstract

Consider the problem setting of Interaction-Grounded Learning (IGL), in which a learner's goal is to optimally interact with the environment with no explicit reward to ground its policies. The agent observes a context vector, takes an action, and receives a feedback vector—using this information to effectively optimize a policy with respect to a latent reward function. Prior analyzed approaches fail when the feedback vector contains the action, which significantly limits IGL's success in many potential scenarios such as Brain-computer interface (BCI) or Human-computer interface (HCI) applications. We address this by creating an algorithm and analysis which allows IGL to work even when the feedback vector contains the action, encoded in any fashion. We provide theoretical guarantees and large-scale experiments based on supervised datasets to demonstrate the effectiveness of the new approach.

## 1  Introduction

Most real-world learning problems, such as BCI and HCI problems, are not tagged with rewards. Consequently, (biological and artificial) learners must infer rewards based on interactions with the environment, which reacts to the learner's actions by generating feedback, but does not provide any explicit reward signal. This paradigm has been previously studied by researchers [e.g., Grizou et al., 2014; Nguyen et al., 2021], including a recent formalization [Xie et al., 2021b] that proposed the term Interaction-Grounded Learning (IGL).

In IGL, the learning algorithm discovers a grounding for the feedback which implicitly discovers a reward function. An information-theoretic impossibility argument indicates additional assumptions are necessary to succeed. Xie et al. [2021b] proceed by assuming the action is conditionally independent of the feedback given the reward. However, this is unnatural in many settings such as neurofeedback in BCI [Katyal et al., 2014; Mishra and Gazzaley, 2015; Debettencourt et al., 2015; Muñoz-Moldes and Cleeremans, 2020; Akinola et al., 2020; Poole and Lee, 2021; Xu et al., 2021] and multimodal interactive feedback in HCI [Pantic and Rothkrantz, 2003; Vitense et al., 2003; Freeman et al., 2017; Mott et al., 2017; Duchowski, 2018; Saran et al., 2018, 2020; Zhang et al.,

---
[*]equal contribution

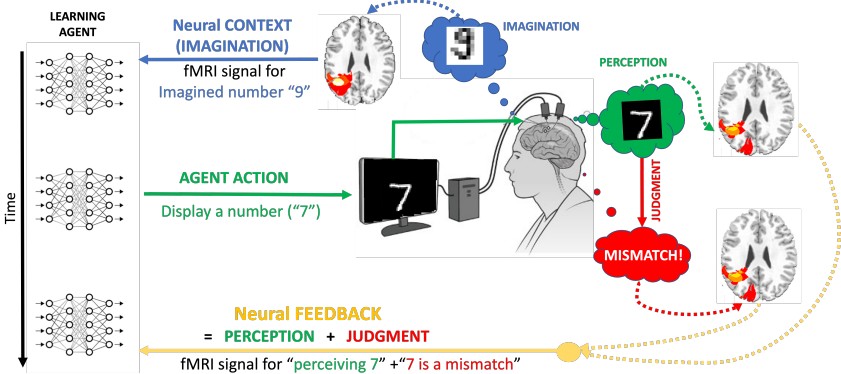

Figure 1: A simulated BCI interaction experiment for a number-guessing task. The human imagines a number, which the agent has to identify through their fMRI signals. The agent displays the number it identifies. The human perceives this number (action) and judges whether it is a match or not (reward)–the mixing of these two activities creates a complex feedback signal, which the learning agent has to decode through the interaction.

2020; Cui et al., 2021; Gao et al., 2021] where the action proceeds and thus influences the feedback. For example, if the feedback is an fMRI recording from a human brain implant in the parietal cortex, and the action of the algorithm is to display a number being imagined by the human (Fig. 1), the parietal cortex feedback will contain information about both perceiving the number being displayed (action) and human's reaction to it for match or mismatch (reward). Another example is determining if a patient in a coma is happy or not, and using their brain signals (feedback) to dissociate the fact that they were asked about being happy (action) and the actual thought that they are happy (reward). An HCI example is a self-calibrating eye tracker used for typing on a screen by an ALS patient [Mott et al., 2017; Gao et al., 2021], where the automated agent calibrates the eye gaze to a certain point on a screen (action) and the human's gaze patterns (feedback) convey a response to the tracker's offset (action) as well as their satisfaction for being able to type a certain key (reward). If you apply the prior IGL approach to such settings, it will fail catastrophically because the requirement of conditional independence is essential to its function. This motivates the question:

*Is it possible to do interaction-ground learning when the feedback has the full information of the action embedded in it?*

We propose a new approach to solve IGL, which we call action-inclusive IGL (AI-IGL), that allows the action to be incorporated into the feedback in arbitrary ways. We consider latent reward as playing the role of latent states, which can be further separated via a contrastive learning method (Section 3.1). Different from the typical latent state discovery in rich-observation reinforcement learning [e.g., Dann et al., 2018; Du et al., 2019; Misra et al., 2020], the IGL setting also requires identifying the semantic meaning of the latent reward states, which is addressed by a symmetry breaking procedure (Section 3.2). We analyze the theoretical properties of the proposed approach, and we prove that it is guaranteed to learn a near-optimal policy as long as the feedback satisfies a weaker context conditional independence assumption. We also empirically evaluate the proposed AI-IGL approach using large-scale experiments on Open-ML's supervised classification datasets [Bischl et al., 2021], as well as a simulated BCI experiment using real human fMRI data [Ellis et al., 2020], demonstrating the effectiveness of the proposed approach (Section 5). Thus, our findings broaden the scope of applicability for IGL.

The paper proceeds as follows. In Section 2, we present the mathematical formulation for IGL. In Section 3, we present a contrastive learning perspective for grounding latent reward which helps to expand the applicability of IGL. In Section 4, we state the resulting algorithm AI-IGL. We provide experimental support for the technique in Section 5 using a diverse set of supervised learning datasets and a simulated BCI dataset. We conclude with discussion in Section 6.

## 2 Background

**Interaction-Grounded Learning** This paper studies the *Interaction-Grounded Learning* (IGL) setting [Xie et al., 2021b], where the learner optimizes for a latent reward by interacting with the

environment and associating ("grounding") observed feedback with the latent reward. At each time step, the learner receives an i.i.d. context $x$ from context set $\mathcal{X}$ and distribution $d_0$. The learner then selects an action $a \in \mathcal{A}$ from a finite action set $|\mathcal{A}| = K$. The environment generates a latent binary reward $r \in \{0, 1\}$ (can be either deterministic or stochastic) and a feedback vector $y \in \mathcal{Y}$ conditional on $(x, a)$, but only the feedback vector $y$ is revealed to the learner. In this paper, we use $R(x, a) := \mathbb{E}_{x,a}[r]$ to denote the expected (latent) reward after executing action $a$ on context $x$. The space of context $\mathcal{X}$ and feedback vector $\mathcal{Y}$ can be arbitrarily large.

Throughout this paper, we use $\pi \in \Pi : \mathcal{X} \to \Delta(\mathcal{A})$ to denote a (stochastic) policy. The expected return of policy $\pi$ is defined by $V(\pi) = \mathbb{E}_{(x,a) \sim d_0 \times \pi}[r]$. The learning goal of IGL is to find the optimal policy in the policy class, $\pi^\star = \arg\max_{\pi \in \Pi} V(\pi)$, only from the observations of context-action-feedback tuples, $(x, a, y)$. This paper mainly considers the batch setting, and we use $\mu$ to denote the behavior policy. In this paper, we also introduce value function classes and decoder classes. We assume the learner has access to a value function class $\mathcal{F}$ where $f \in \mathcal{F} : \mathcal{X} \times \mathcal{A} \to [0, 1]$ and reward decoder class $\psi \in \Psi : \mathcal{Y} \times \mathcal{A} \to [0, 1]$. We defer the assumptions we made on these classes to Section 3 for clarity.

We may hope to solve IGL without any additional assumptions. However, it is information-theoretically impossible without additional assumptions, even if the latent reward is decodable from $(x, a, y)$, as demonstrated by the following example.

**Example 1** (Hardness of assumption-free IGL). *Suppose $y = (x, a)$ and suppose the reward is deterministic in $(x, a)$. In this case, the latent reward $r$ can be perfectly decoded from $y$. However, the learner receives no more information than $(x, a)$ from the $(x, a, y)$ tuple. Thus if $\Pi$ contains at least 2 policies, for any environment where any IGL algorithm succeeds, we can construct another environment with the same observable statistics where that algorithm must fail.*

**IGL with full conditional independence** Example 1 demonstrates the need for further assumptions to succeed at IGL. Xie et al. [2021b] proposed an algorithm that leverages the following conditional independence assumption to facilitate grounding the feedback in the latent reward.

**Assumption 1** (Full conditional independence). *For arbitrary $(x, a, r, y)$ tuples where $r$ and $y$ are generated conditional on the context $x$ and action $a$, we assume the feedback vector $y$ is conditionally independent of context $x$ and action $a$ given the latent reward $r$, i.e. $x, a \perp\!\!\!\perp y | r$.*

Xie et al. [2021b] introduce a reward decoder class $\psi \in \Psi : \mathcal{Y} \to [0, 1]$ for estimating $\mathbb{E}[r|y]$, which leads to the decoded return $V(\pi, \psi) := \mathbb{E}_{(x,a) \sim d_0 \times \pi}[\psi(y)]$. They proved that it is possible to learn the best $\pi$ and $\psi$ jointly by optimizing the following proxy learning objective:

$$\arg\max_{(\pi, \psi) \in \Pi \times \Psi} \mathcal{J}(\pi, \psi) := V(\pi, \psi) - V(\pi_{\mathsf{base}}, \psi), \tag{1}$$

where $\pi_{\mathsf{base}}$ is a policy known to have low expected return. Over this paper, we use IGL (full CI) to denote the proposed algorithm of Xie et al. [2021b].

## 3 A Contrastive-learning Perspective for Grounding Latent Reward

The existing work of interaction-grounded learning leverages the assumption of full conditional independence (Assumption 1), where the feedback vector only contains information from the latent reward. The goal of this paper is to relax these constraining assumptions and broaden the scope of applicability for IGL. This paper focuses on the scenario where the action information is possibly embedded in the feedback vector, which is formalized by the following assumption.

**Assumption 2** (Context Conditional Independence). *For arbitrary $(x, a, r, y)$ tuple where $r$ and $y$ are generated conditional on the context $x$ and action $a$, we assume the feedback vector $y$ is conditionally independent of context $x$ given the latent reward $r$ and action $a$. That is, $x \perp\!\!\!\perp y | a, r$.*

Assumption 2 allows the feedback vector $y$ to be generated from both latent reward $r$ and action $a$, differing from Assumption 1 which constrains the feedback vector $y$ to be generated based only on the latent reward $r$. We discuss the implications at the end of this section.

## 3.1 Grounding Latent Reward via Contrastive Learning

In this section, we propose a contrastive learning objective for interaction-grounded learning, which further guides the design of our algorithm. We perform derivations with exact expectations for clarity and intuitions and provide finite-sample guarantees in Section 4.

**Assumption 3** (Separability). *For each $\bar{a} \in \mathcal{A}$, there exists an $(f_{\bar{a}}^{\star}, \psi_{\bar{a}}^{\star}) \in \mathcal{F} \times \Psi$, such that: 1) $\mathbb{E}[\psi_{\bar{a}}^{\star}(y, \bar{a}) | \bar{a}, r = 1] - \mathbb{E}[\psi_{\bar{a}}^{\star}(y, \bar{a}) | \bar{a}, r = 0] = 1$; 2) $|\mathbb{E}_{\mu}[f_{\bar{a}}^{\star}(x, \bar{a}) | \bar{a}, r = 1] - \mathbb{E}_{\mu}[f_{\bar{a}}^{\star}(x, \bar{a}) | \bar{a}, r = 0]| \geq \Delta_{\mathcal{F}}$. We also assume $1 - \Psi \subseteq \Psi$, where $1 - \Psi := \{1 - \psi(\cdot, \cdot) : \psi \in \Psi\}$.*

Assumption 3 consists of two components. For $\Psi$, it is a realizability assumption that ensures that a perfect reward decoder is included in the function classes. Although this superficially appears unreasonably strong, note $y$ is generated based upon $a$ and the *realization* of $r$; therefore this is compatible with stochastic rewards. For $\mathcal{F}$, it ensures the expected predicted reward conditioned on the latent reward, having value $r \in \{0, 1\}$ and the action being $a$, is separable. When $\mu = \pi_{\mathsf{unif}}$, $\Delta_{\mathcal{F}}$ can be lower bounded by $\max_{f \in \mathcal{F}} 4|\mathrm{Cov}_{\pi_{\bar{a}}}(f, R)|$ ($\pi_{\bar{a}}$ denotes the constant policy with action $\bar{a}$). Detailed proof of this argument can be found in Appendix A. One sufficient condition is that the expected latent reward $R(\cdot, \bar{a})$ has enough variance and $R \in \mathcal{F}$. The condition of $1 - \Psi \subseteq \Psi$ can be constructed easily via standard realizable classes. That is, if $\psi_{\bar{a}}^{\star} \in \Psi'$ for some classes $\Psi'$, simply setting $\Psi \leftarrow \Psi' \bigcup (1 - \Psi')$ satisfies Assumption 3. Note that, this construction of $\Psi$ only amplifies the size of $\Psi'$ by a factor of 2.

**Reward Prediction via Contrastive Learning**   We now formulate the following contrastive-learning objective for solving IGL. Suppose $\mu(a|x)$ is the behavior policy. We also abuse $\mu(x, a, y)$ to denote the data distribution, and $\mu_a(x, y), \mu_a(x), \mu_a(y)$ to denote the marginal distribution under action $a \in \mathcal{A}$. We construct an augmented data distribution for each $a \in \mathcal{A}$: $\widetilde{\mu}_a(x, y) := \mu_a(x) \cdot \mu_a(y)$ (i.e., sampling $x$ and $y$ independently from $\mu_a$).

Conceptually, for each action $a \in \mathcal{A}$, we consider tuples $(x, y) \sim \mu_a(x, y)$ and $(\widetilde{x}, \widetilde{y}) \sim \widetilde{\mu}_a(x, y)$. From Assumption 3, conditioned on $(x, a)$ the optimal feedback decoder $\psi^{\star}(y, a)$ has mean equal to the optimal reward predictor $f^{\star}(x, a)$. Therefore we might seek an $(f, \psi)$ pair which minimizes any consistent loss function, e.g., squared loss. However this is trivially achievable, e.g., by having both always predict 0. Therefore we formulate a contrastive-learning objective, where we maximize the loss between the predictor and decoder on the augmented data distribution. For each $a \in \mathcal{A}$, we solve the following objective

$$\operatorname*{argmin}_{(f_a, \psi_a) \in \mathcal{F} \times \Psi} \mathcal{L}_a(f_a, \psi_a) := \mathbb{E}_{\mu_a}\left[(f_a(x, a) - \psi_a(y, a))^2\right] - \mathbb{E}_{\widetilde{\mu}_a}\left[(f_a(\widetilde{x}, a) - \psi_a(\widetilde{y}, a))^2\right]. \quad (2)$$

In the notation of equation (2), the $a$ subscript indicates the $(f_a, \psi_a)$ pair are optimal for action $a$. Note they are always evaluated at $a$, which we retain as an input for compatibility with the original function classes.

Note that $\mathcal{L}$ is also similar to many popular contrastive losses [e.g., Wu et al., 2018; Chen et al., 2020] especially for the spectral contrastive loss [HaoChen et al., 2021].

$$
\begin{aligned}
\mathcal{L}_a(f_a, \psi_a) &= \mathbb{E}_{\mu_a}\left[f_a(x, a)^2 - 2f_a(x, a)\psi_a(y, a) + \psi_a(y, a)^2\right] \\
&\quad - \mathbb{E}_{\widetilde{\mu}_a}\left[f_a(x, a)^2 - 2f_a(x, a)\psi_a(y, a) + \psi_a(y, a)^2\right] \\
&= -2\left(\mathbb{E}_{\mu_a}[f_a(x, a)\psi_a(y, a)] - \mathbb{E}_{\widetilde{\mu}_a}[f_a(x, a)\psi_a(y, a)]\right) \quad \textbf{(spectral contrastive loss)} \\
&= -2\left(\mathbb{E}_{\mu_a}[f_a(x, a)\psi_a(y, a)] - \mathbb{E}_{\mu_a}[f_a(x, a)]\mathbb{E}_{\mu_a}[\psi_a(y, a)]\right). \\
&\hspace{7cm} (\widetilde{\mu}_a(x, y) = \mu_a(x) \cdot \mu_a(y))
\end{aligned}
$$

Below we show that minimizing $\mathcal{L}_a(f_a, \psi_a)$ decodes the latent reward under Assumptions 2 and 3 up to a sign ambiguity. For simplicity, we introduce the notation of $f_{a,r}$ and $\psi_{a,r}$ for any $(f, \psi) \in \mathcal{F} \times \Psi$,

$$f_{a,r} := \sum_{x} \Pr(x|a, r) f(x, a), \quad \psi_{a,r} := \sum_{y} \Pr(y|a, r)\psi(y, a). \quad (3)$$

$f_{a,r}$ and $\psi_{a,r}$ are the expected predicted reward of $f(x, a)$ and decoded reward $\psi(y, a)$ (under the behavior policy $\mu$ for $f_{a,r}$) conditioned on the latent reward having value $r$ and the action being $a$.

**Proposition 1.** *For any action $a \in \mathcal{A}$, if $\mu(a|x) > 0$ for all $x \in \mathcal{X}$ and Assumption 2 and 3 hold, and let $(\widehat{f}_a, \widehat{\psi}_a)$ be the solution of Eq.(2). Then, $|\widehat{f}_{a,1} - \widehat{f}_{a,0}| = \max_{f \in \mathcal{F}} |f_{a,1} - f_{a,0}|$ and $|\widehat{\psi}_{a,1} - \widehat{\psi}_{a,0}| = \max_{\psi \in \Psi} |\psi_{a,1} - \psi_{a,0}|$.*

*Proof Sketch.* For any policy $\pi$, we use $d_a^\pi := \sum_x d_0(x)\pi(a|x)$ to denote the visitation occupancy of action $a$ under policy $\pi$, and $\rho_a^\pi := \frac{1}{d_a^\pi}\sum_x d_0(x)\pi(a|x)R(x,a)$ to denote the average reward received under executing action $a$. Then, by the context conditional independent assumption $x \perp\!\!\!\perp y | r, a$ (Assumption 2), we know

$$\mathbb{E}_{\mu_a}[f(x)\psi(y)] = (1-\rho_a^\mu)f_{a,0}\psi_{a,0} + \rho_a^\mu f_{a,1}\psi_{a,1}$$

$$\mathbb{E}_{\mu_a}[f(x)]\mathbb{E}_{\mu_a}[\psi(y)] = (1-\rho_a^\mu)^2 f_{a,0}\psi_{a,0} + (\rho_a^\mu)^2 f_{a,1}\psi_{a,1} + (1-\rho_a^\mu)\rho_a^\mu(f_{a,0}\psi_{a,1} + f_{a,1}\psi_{a,0})$$

$$\implies \mathcal{L}_a(f_a,\psi_a) \propto -(f_{a,1}-f_{a,0})(\psi_{a,1}-\psi_{a,0}).$$

Therefore, separately maximizing $|f_{a,1}-f_{a,0}|$ and $|\psi_{a,1}-\psi_{a,0}|$ maximizes $\mathcal{L}_a(f_a,\psi_a)$. □

### 3.2 Symmetry Breaking

In the last section, we demonstrated that the latent reward could be decoded in a contrastive-learning manner up to a sign ambiguity. The following example demonstrates the ambiguity.

**Example 2** (Why do we need extra information to identify the latent reward?). *In the optimal solution of objective Eq.(2), both $\widehat{\psi}_a$ and $\widehat{\psi}'_a := 1 - \widehat{\psi}_a$ yield the same value. It is information-theoretically difficult to distinguish which one of them is the correct solution without extra information. That is because, for any environment* ENV1*, there always exists a "symmetric" environment* ENV2*, where: 1) $R(x,a)$ of* ENV1 *is identical to $(1-R(x,a))$ of* ENV2 *for all $(x,a) \in \mathcal{X} \times \mathcal{A}$; 2) the the conditional distribution of $y|r,a$ in* ENV1 *is identical to the conditional distribution of $y|1-r,a$ in the* ENV2 *for all $a \in \mathcal{A}$. In this example,* ENV1 *and* ENV2 *will always generate the identical distribution of feedback vector $y$ after any $(x,a) \in \mathcal{X} \times \mathcal{A}$. However,* ENV1 *and* ENV2 *have the exactly opposite latent reward information.*

As we demonstrate in Example 2, the learned decoder from Eq.(2) could be corresponding to a symmetric pair of semantic meanings, and identifying them without extra information is information-theoretically impossible. The *symmetry breaking* procedure is one of the key challenges of interaction-grounded learning. To achieve symmetry breaking, we make the following assumption to ensure the identifiability of the latent reward.

**Assumption 4** (Baseline Policies). *For each $a \in \mathcal{A}$, there exists a baseline policy $\pi_{\mathsf{base}}^a$, such that,*

(a) $\pi_{\mathsf{base}}^a$ *satisfies* $\sum_x d_0(x)\pi(a|x) \geq c_m > 0$.

(b) $|\frac{1}{2} - \rho_a^{\pi_{\mathsf{base}}^a}| \geq \eta$, *where* $\rho_a^{\pi_{\mathsf{base}}^a} = \frac{\sum_x d_0(x)\pi_{\mathsf{base}}^a(a|x)R(x,a)}{\sum_x d_0(x)\pi_{\mathsf{base}}^a(a|x)}$.

To instantiate Assumption 4 in practice, we provide the following simple example of $\pi_{\mathsf{base}}$ that satisfies Assumption 4. Suppose $\pi_{\mathsf{base}} = \pi_{\mathsf{unif}}$ (uniformly random policy), and we have "all constant policies are bad", i.e., $V(\pi_{\bar{a}} = \mathbb{1}(a = \bar{a})) < \frac{1}{2} - \eta$ for all $\bar{a} \in \mathcal{A}$. Then it is easy to verify that $c_m = \frac{1}{K}$ and $\rho_{\bar{a}}^{\pi_{\mathsf{base}}} \leq \frac{1}{2} - \eta$ for all $\bar{a} \in \mathcal{A}$.

Note that $\pi_{\mathsf{base}}^a$ can be different over actions. Intuitively, Assumption 4(a) is saying that the total probability of $\pi_{\mathsf{base}}^a$ selecting action $a$ (over all context $x \in \mathcal{X}$) is at least $c_m$. This condition ensures that $\pi_{\mathsf{base}}^a$ has enough visitation to action $a$ and makes symmetry breaking possible. Assumption 4(b) states that if we only consider the reward obtained from taking action $a$, $\pi_{\mathsf{base}}^a$ is known to be either "sufficiently bad" or "sufficiently good". Note the directionality of the extremeness of $\pi_{\mathsf{base}}^a$ must be known, e.g., a policy which has a unknown reward of either 0 or 1 is not usable. This condition follows a similar intuition as the identifiability assumption of Xie et al. [2021b, Assumption 2] and breaks the symmetry. For example, consider the ENV1 and ENV2 introduced in Example 2, $\rho_{a,\mathsf{ENV1}}^\pi = 1 - \rho_{a,\mathsf{ENV2}}^\pi$ for any policy $\pi$. To separating ENV1 and ENV2 using some policy $\pi$, $\rho_{a,\mathsf{ENV1}}^\pi$ and $\rho_{a,\mathsf{ENV2}}^\pi$ require to have a non-zero gap, which leads to Assumption 4(b).

The effectiveness of symmetry breaking under Assumption 4 can be summarized as below: we conduct the following estimation of $\rho_a^{\pi_{\mathsf{base}}}$, using the learned $\widehat{\psi}_a$, $\widehat{\rho}_a^{\pi_{\mathsf{base}}} = \frac{\sum_x d_0(x)\pi_{\mathsf{base}}^a(a|x)\widehat{\psi}_a(x,a)}{\sum_x d_0(x)\pi_{\mathsf{base}}^a(a|x)}$. If $\widehat{\psi}_a$ can efficiently decode the latent reward, then $\widehat{\rho}_a^{\pi_{\mathsf{base}}^a}$ converges to either $\rho_a^{\pi_{\mathsf{base}}}$ or $1 - \rho_a^{\pi_{\mathsf{base}}}$. Therefore, applying Assumption 4(b) breaks the symmetry.

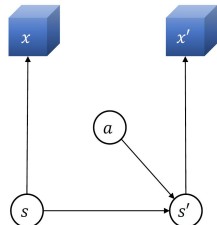 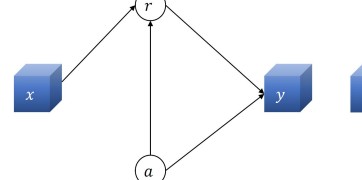 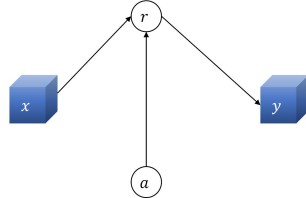

(a) Latent State in Rich-Observation RL [Misra et al., 2020]

(b) Latent Reward in IGL under Assumption 2 (this paper)

(c) Latent Reward in IGL under Assumption 1 [Xie et al., 2021b]

Figure 2: Causal graphs of interaction-grounded learning under different assumptions as well as rich-observation reinforcement Learning.

### 3.3 Comparison to Full CI

When we have the context conditional independence, it is easy to verify the failure of optimizing the original IGL objective Eq.(1) by the following example.

**Example 3** (Failure of the original IGL objective under Assumption 2). *Let $\mathcal{X} = \mathcal{A} = \{1, 2, \ldots, 10\}$ and feedback vector is generated by $y = (a + R(x, a)) \mod 10$ (we use % to denote $\mod$ in the following part). We also assume $d_0(x) = \pi_{\text{base}}(a|x) = 1/10$ for any $(x, a) \in \mathcal{X} \times \mathcal{A}$ and $R(x, a) = \pi^\star(a|x) := \mathbb{1}(x = a)$. Then, we have, for any $\psi : \mathcal{Y} \to [0, 1]$ (approach proposed by Xie et al. [2021b] assumes the reward decoder only takes feedback vector $y$ as the input),*

$$\mathcal{L}(\pi^\star, \psi) = \frac{1}{10} \sum_{x=1}^{10} \sum_{a=1}^{10} \mathbb{1}(x = a)\psi((a+1)\%10) - \frac{1}{100} \sum_{x=1}^{10} \sum_{a=1}^{10} \psi((a + \mathbb{1}(x = a))\%10)$$
$$= \frac{1}{10} \sum_{a=1}^{10} \psi(a) - \frac{1}{10} \sum_{a=1}^{10} \psi(a) = 0.$$

*On the other hand, consider the constant policy $\pi_1(a|x) := \mathbb{1}(a = 1)$ for all $x \in \mathcal{X}$ and decoder $\psi_2(y) := \mathbb{1}(y = 2)$ for all $y \in \mathcal{Y}$, then,*

$$\mathcal{L}(\pi_1, \psi_2) = \frac{1}{10} \sum_{x=1}^{10} \sum_{a=1}^{10} \mathbb{1}(a = 1)\psi_2((a+1)\%10) - \frac{1}{100} \sum_{x=1}^{10} \sum_{a=1}^{10} \psi_2((a + \mathbb{1}(x = a))\%10)$$
$$= \psi_2(2) - \frac{1}{10} \sum_{a=1}^{10} \psi_2(a) = 0.9 > \mathcal{L}(\pi^\star, \psi), \quad \forall \psi \in \Psi.$$

*This implies that maximizing the original IGL objective Eq.(1) could not always converge to $\pi^\star$ when we only have the context conditional independence.*

This example indicates optimizing a combined contrastive objective with a single symmetry-breaking policy is insufficient to succeed in our $x \perp\!\!\!\perp y | r, a$ case. Our current approach corresponds to optimizing a contrastive objective and breaking symmetry for each action separately rather than simultaneously.

### 3.4 Viewing Latent Reward as a Latent State

Our approach is motivated by latent state discovery in Rich-Observation RL [Misra et al., 2020]. Figure 2 compares the causal graphs of Rich-Observation RL, IGL with context conditional independence, and IGL with full conditional independence. In Rich-Observation RL, a contrastive learning objective is used to discover latent states; whereas in IGL a contrastive learning objective is used to discover latent rewards. In this manner we view latent rewards analogously to latent states.

Identifying latent states up to a permutation is completely acceptable in Rich-Observations RL, as the resulting imputed MDP orders policies identically to the true MDP. However in IGL the latent states have scalar values associated with them (rewards), and identifying up to a permutation is not sufficient to order policies correctly. Thus, we require the additional step of symmetry breaking.

## 4 Main Algorithm

This section instantiates the algorithm for IGL with action-inclusive feedback, following the concept we introduced in Section 3. For simplicity, we select the uniformly random policy $\pi_{\text{unif}}$ as behavior policy $\mu$ in this section. That choice of $\mu$ can be further relaxed using an appropriate importance weight. We first introduce the empirical estimation of $\mathcal{L}_a$ as follows ($\mathcal{L}_a$ is defined in Eq.(2). $\mathcal{J}_{\bar{a},\mathcal{D}}$

estimates the spectral contrastive loss, which corresponds to $-\mathcal{L}_a$). For any $\bar{a} \in \mathcal{A}$, we define the following empirical estimation of the spectral contrastive loss:

$$\mathcal{J}_{\bar{a},\mathcal{D}}(f,\psi) = \mathbb{E}_{\mathcal{D}}\left[f(x,a)\psi(y,a)\mathbb{1}(a=\bar{a})\right] - \mathbb{E}_{\mathcal{D}}\left[f(x,a)\mathbb{1}(a=\bar{a})\right]\mathbb{E}_{\mathcal{D}}\left[\psi(y,a)\mathbb{1}(a=\bar{a})\right]. \quad (4)$$

Using this definition, Algorithm 1 instantiates a version of the IGL algorithm with action-inclusive feedback. Without loss of generality, we also assume $\rho_a^{\pi_{\text{base}}^a} \leq 1/2$ for Assumption 4(b) for all $a \in \mathcal{A}$ in this section. The case of $\rho_a^{\pi_{\text{base}}^a} > 1/2$ for some action $a$ can be addressed by modifying the symmetry-breaking step properly in Algorithm 1.

---

**Algorithm 1** Action-inclusive IGL (AI-IGL)

---

**Input:** Batch data $\mathcal{D}$ generated by $\mu = \pi_{\text{unif}}$. baseline policy $\pi_{\text{base}}^{a \in \mathcal{A}}$.

1: Initialize policy $\pi_1$ as the uniform policy.
2: **for** $\bar{a} \in \mathcal{A}$ **do**
3:     Obtain $(f_{\bar{a}}, \psi_{\bar{a}})$ by                                      ▷ *Latent State (Reward) Discovery*

$$(f_{\bar{a}}, \psi_{\bar{a}}) \leftarrow \operatorname*{argmax}_{(f,\psi) \in \mathcal{F} \times \Psi} \mathcal{J}_{\bar{a},\mathcal{D}}(f,\psi), \quad (5)$$

    where $\mathcal{J}_{\bar{a},\mathcal{D}}(f,\psi)$ is defined in Eq.(4)
4:     Compute $\widehat{\varrho}_{\bar{a}}^{\pi_{\text{base}}}$ by $\widehat{\varrho}_{\bar{a}}^{\pi_{\text{base}}} = \frac{\sum_{(x,a,y) \in \mathcal{D}} \pi_{\text{base}}^{\bar{a}}(a|x)\psi_{\bar{a}}(x,\bar{a})\mathbb{1}(a=\bar{a})}{\sum_{(x,a,y) \in \mathcal{D}} \pi_{\text{base}}^{\bar{a}}(a|x)\mathbb{1}(a=\bar{a})}$.      ▷ *Symmetry Breaking*
5:     **if** $\widehat{\varrho}_{\bar{a}}^{\pi_{\text{base}}} > \frac{1}{2}$ **then**    $\psi'_{\bar{a}} \leftarrow (1 - \psi_{\bar{a}})$.
6:     **else**   $\psi'_{\bar{a}} \leftarrow \psi_{\bar{a}}$.
7:     **end if**
8: **end for**
9: Generate decoded contextual bandits dataset $\mathcal{D}_{\text{CB}} \leftarrow \{(x, a, \psi'_a(y,a), \mu(a|x)) : (x,a,y) \in \mathcal{D}\}$.
10: Output policy $\widehat{\pi}(x) \leftarrow \text{CB}(\mathcal{D}_{\text{CB}})$, where CB denotes an offline contextual bandit oracle.

---

At a high level, Algorithm 1 has two separate components, latent state (reward) discovery (line 3) and symmetry breaking (line 4-7), for each action in $\mathcal{A}$.

**Theoretical guarantees**   In Algorithm 1, the output policy $\widehat{\pi}$ is obtained by calling an offline contextual bandits oracle (CB). We now formally define this oracle and its expected property.

**Definition 1** (Offline contextual bandits oracle). *An algorithm CB is called an offline contextual bandit oracle if for any dataset* $\mathcal{D} = \{(x_i, a_i, r_i, \mu(a_i|x_i))\}_{i=1}^{|\mathcal{D}|}$ ($x_i \sim d_0, a_i \sim \mu$, and $r_i$ is the reward determined by $(x_i, a_i)$) and any policy class $\Pi$, the policy $\widehat{\pi}$ produced by $\text{CB}(\mathcal{D})$ satisfies $\varepsilon_{\text{CB}} := \max_{\pi \in \Pi} \mathbb{E}_{d_0 \times \pi}[r] - \mathbb{E}_{d_0 \times \widehat{\pi}}[r] \leq o(1)$.

The notion in Definition 1 corresponds to the standard policy learning approaches in the contextual bandits literature [e.g., Langford and Zhang, 2007; Dudik et al., 2011; Agarwal et al., 2014], and typically leads to $\varepsilon_{\text{CB}} = \sqrt{K^{\log |\Pi|/\delta}/|\mathcal{D}|}$. We now provide the theoretical analysis of Algorithm 1. In this paper, we use $d_{\mathcal{F},\Psi}$ to denote the joint statistical complexity of the class of $\mathcal{F}$ and $\Psi$. For example, if the function classes are finite, we have $d_{\mathcal{F},\Psi} = \mathcal{O}(\log |\mathcal{F}||\Psi|/\delta)$, and $\delta$ is the failure probability. The infinite function classes can be addressed by some advanced methods such as covering number or Rademacher complexity [see, e.g., Mohri et al., 2018]. The following theorem provides the performance guarantee of the output policy of Algorithm 1.

**Theorem 2.** *Suppose Assumptions 2, 3 and 4 hold, $\Delta_{\mathcal{F}}$ is defined in Assumption 3, and $\widehat{\pi}$ be the output policy of Algorithm 1. If we have* $|\mathcal{D}| \geq \mathcal{O}\left(\frac{K^3 d_{\mathcal{F},\Psi}}{(\min\{\eta\Delta_{\mathcal{F}}, Kc_m\})^2}\right)$, *then, with high probability,*

$$V(\pi^\star) - V(\pi) \leq \mathcal{O}\left(\frac{1}{\Delta_{\mathcal{F}}}\sqrt{\frac{K^3 d_{\mathcal{F},\Psi}}{|\mathcal{D}|}}\right) + \varepsilon_{\text{CB}}.$$

Similar to the performance of Xie et al. [2021b], the learner is guaranteed to converge in the right direction only after we have sufficient data for symmetry breaking. The dependence on $K$ in Theorem 2 can be improved as different action has a separate learning procedure. For example, if we consider $\mathcal{F} = \mathcal{F}_1 \times \mathcal{F}_2 \times \cdots \times \mathcal{F}_K$ and $\Psi = \Psi_1 \times \Psi_2 \times \cdots \times \Psi_K$, where $\mathcal{F}_a$ and $\Psi_a$ are independent components that is only corresponding to action $a$ (this is a common setup for linear approximated

reinforcement learning approaches with discrete action space). If $\mathcal{F}_a$ and $\Psi_a$ are identical copies of $K$ separate classes, we know $\log|\mathcal{F}| = K\log|\mathcal{F}_a|$ and $\log|\Psi| = K\log|\Psi_a|$, which leads a $\sqrt{K}$ improvement.

We now provide the proof sketch of Theorem 2 and we defer the detailed proof to Appendix A.

*Proof Sketch.* The proof of Theorem 2 consists of two different components—discovering latent reward and breaking the symmetry, which are formalized by the following lemma.

**Lemma 3** (Discovering latent reward). *Suppose Assumptions 2 and 3 hold, and let $(f_{\bar{a}}, \psi_{\bar{a}})$ be obtained by Eq.(5). Then, with high probability, we have $|\psi_{\bar{a},1} - \psi_{\bar{a},0}| \geq \left(1 - \mathcal{O}\left(\frac{1}{\Delta_\mathcal{F}}\sqrt{\frac{K^3 d_{\mathcal{F},\Psi}}{|\mathcal{D}|}}\right)\right)$.*

Lemma 3 ensures that the learned decoder on Eq.(5) correctly separates the latent reward. In particular since $\psi$ ranges over $[0, 1]$, Lemma 3 ensures $\max\left(\Pr(\psi(y, a) = r), \Pr(\psi(y, a) = 1 - r)\right) > 1 - o(1)$ under the behaviour policy. Thus, if we can break symmetry, we can use $\psi$ to generate a reward signal and reduce to ordinary contextual bandit learning. The following lemma guarantees the correctness of the symmetry-breaking step.

**Lemma 4** (Breaking symmetry). *Suppose Assumption 4 holds. For any $\bar{a} \in \mathcal{A}$, if we have $|\mathcal{D}| \geq \mathcal{O}\left(\frac{K^3 d_{\mathcal{F},\Psi}}{(\min\{\eta\Delta_\mathcal{F}, Kc_m\})^2}\right)$, then, with high probability, $\psi'_{\bar{a},1} \geq \psi'_{\bar{a},0}$.*

Combining these two lemmas above as well as the CB oracle establishes the proof of Theorem 2, and the detailed proof can be found in Appendix A. $\qquad\square$

# 5 Empirical Evaluations

In this section, we provide empirical evaluations in simulated environments created using supervised classification datasets (Sec. 5.1) and an open source fMRI simulator (Sec. 5.2). We evaluate our approach by comparing: (1) CB: Contextual Bandits with exact reward, (2) IGL (full CI): The method proposed by Xie et al. [2021b] which assumes the feedback vector contains no information about the context and action, and (3) AI-IGL: The proposed method which assumes that the feedback vector could contain information about the action but is conditionally independent of the context given the reward. Note that contextual bandits (CB) is a skyline compared to both IGL (full CI) and AI-IGL, since it need not disambiguate the latent reward. All methods use logistic regression with a linear representation. At test time, each method takes the argmax of the policy. Following the practical instantiation of Assumption 4 (Section 3.2), we know if the dataset has a balanced action distribution (no action belongs to more than 50% of the samples), selecting uniformly random policy $\pi_{\mathsf{unif}}$ as $\pi_{\mathsf{base}}^a$ for all $a \in \mathcal{A}$ satisfies Assumption 4. Therefore, in this section, our experiments are all based on the dataset with a balanced action distribution, and we select $\pi_{\mathsf{base}}^a = \pi_{\mathsf{unif}}$ for all $a \in \mathcal{A}$.

## 5.1 Large-scale Experiments with OpenML CC-18 Datasets

To verify that our proposed algorithm scales to a variety of tasks, we evaluate performance on more than 200 datasets from the publicly available OpenML Curated Classification Benchmarking Suite [Vanschoren et al., 2015; Casalicchio et al., 2019; Feurer et al., 2021; Bischl et al., 2021]. OpenML CC-18 datasets are licensed under CC-BY license[2] and the platform and library are licensed under the BSD (3-Clause) license[3]. At each time step, the context $x_t$ is generated uniformly at random. The learner selects an action $a_t \in \{0, \ldots, K-1\}$ as the predicted label of $x_t$ (where $K$ is the total number of actions available in the environment). The binary reward $r_t$ is the correctness of the prediction label $a_t$. The feedback $y_t$ is a two dimensional vector $(a_t, r_t)$. Each dataset has a different sample size $(N)$ and a different set of available actions $(K)$. We sample datasets with 3 or more actions, and with a balanced action distribution (no action belongs to more than 50% of the samples) to satisfy Assumption 4. We use 90% of the data for training and the remaining 10% for evaluation. Additional details of setting up the experiment are in Appendix C. The results are averaged over 20 trials and shown in Table 1.

[2] https://creativecommons.org/licenses/by/4.0/
[3] https://opensource.org/licenses/BSD-3-Clause

| Dataset Criteria | Dataset Count | Constant Action | CB Policy Accuracy (%) | IGL (full CI) Policy Accuracy (%) | Performance w.r.t CB | AI-IGL Policy Accuracy (%) | Performance w.r.t CB |
|---|---|---|---|---|---|---|---|
| $K \geq 3$ | 271 | $25.28 \pm 2.96$ | $57.98 \pm 5.65$ | $15.65 \pm 2.30$ | $0.30 \pm 0.04$ | $35.74 \pm 1.45$ | $0.59 \pm 0.02$ |
| $K \geq, N \geq 70000$ | 83 | $22.57 \pm 3.14$ | $58.41 \pm 5.04$ | $11.91 \pm 1.80$ | $0.22 \pm 0.04$ | $50.11 \pm 2.98$ | $0.79 \pm 0.03$ |

Table 1: Results in the OpenML environments with two-dimensional action-inclusive feedback. Average and standard error reported over 20 trials for each algorithm. 'Performance w.r.t. CB' reports the ratio of an IGL method's policy accuracy over CB policy accuracy.

## 5.2 Experiments on Realistic fMRI Simulation

We also evaluate our approach on a simulated BCI experiment, where a human imagines a number and the agent faces a number-decoding task (Fig. 1). We simulate an fMRI based BCI setup, recording the fMRI BOLD (Blood Oxygenation Level Dependent) response in the parietal cortex of a human brain. We have selected the parietal cortex since both numerical processing [Santens et al., 2010] and match-mismatch processing are reported to involve this region [Male et al., 2020], and in principle it could feasibly be recorded by fNIRS (with similar spatial resolution to fMRI) [Naseer and Hong, 2015]. The task requires an intelligent agent to decode the numbers the human imagines. This task is motivated by real-world examples where an implant could assist a person, user, or patient by communicating their mental states to the outside world, whether for therapeutic purposes or to control hardware or software.

The task setup consists of the following steps—(1) Human Imagination: A human participant imagines a number from a set of digits (i.e., $7, 8, 9$), (2) Agent Decoding: an agent (Full CI IGL or AI-IGL) decodes the number being imagined using only the fMRI signals of the human participant, with no labels and no supervised pretraining, (3) Human Perception: the agent visually displays the number it decoded to the human, which the human brain perceives, and hence the fMRI signals would contain patterns related to the perception of the agent's guess, (4) Human Judgment: Once the human judges whether the agent-guessed number yields a match or a mismatch to the number they imagined, fMRI signals reflect the participant's judgment. This simple setup mimics many relevant applications of BCI with a realistic simulation of fMRI signals (generalizable to the more cost-efficient fNIRS). We used a third party simulation software, offering a strong test bed for the algorithms compared here.

We use an open source simulator [Ellis et al., 2020] to simulate BCI-like, interactive fMRI signals for the imagination, perception, and judgment phases described above. This simulator leverages real human fMRI data [Bejjanki et al., 2017] to simulate signals for different regions of the brain. Prior neuroscience studies have shown that the parietal cortices in the human brain (in particular the posterior parietal cortex) process numbers [Santens et al., 2010], while the frontoparietal cortices process visual attention as well as match-mismatch judgments [Male et al., 2020]. These studies, together with the low temporal resolution of fMRI/fNIRS [Glover, 2011], warrant the assumption we make here: that number perception and judgments signals can both be decoded from the same region, i.e., the parietal cortex. Therefore, per AI-IGL's theoretical assumptions, the two signals can be mixed in the brain's implicit feedback signal, here simulated as an fMRI or fNIRS signal, in arbitrary ways. Thus, we selected a region of interest (ROI) in the parietal cortex to cover the mixed feedback of perceiving the numbers and making a match-mismatch judgments. This simulated BCI setting is precisely the sort of realistic condition that our proposed AI-IGL solution is designed to handle well. Moreover, it is quite possible to conduct this experiment in principle using fMRI, or more realistically using fNIRS–which is portable, noninvasive, easy to use, and can be focused on a specific patch on the surface of the brain.

Each event (perception, imagination, judgment) evokes different activity in the human brain, within the same set of $4 \times 4 \times 4$ voxels, generating a 64-dimensional activity vector. We generate a time course of 666 trials, each of them including a sequence of events where an imagination event is followed by a response from the agent, then a perception event, and a judgment event. Thus, the 666 trials include 1998 simulated brain events total. Each event is 2 seconds long, with an inter-stimulus (ISI) of 7 seconds, and a temporal resolution of 10 seconds. For each event, the simulator combines spatiotemporal noise (trial noise) and voxel activation patterns to simulate activity signals. We use the imagination signal as context (Fig. 1, Context) and a weighted average of the perception and judgement signals to simulate the brain's mixed feedback signal (Fig. 1, Feedback).

We evaluate the performance of both full CI IGL and AI-IGL on this simulated and interactive number-decoding task, under varying levels of feedback noise. The results are shown in Table 2. We

found that full CI IGL failed at the task, remaining at chance level guessing. However, the novel algorithm proposed here, AI-IGL, was capable of learning to dissociate the human judgment from a mixed feedback signal. The significance of these findings are two-fold. First, they offer a realistic approach to simulating BCI experiments in order to test theoretical advances before conducting real world experiments. Second, our results further validate the theoretical advances of AI-IGL compared to the previous IGL approach for a realistic interactive brain-computer interface.

| Algorithm | 1% noise | 5% noise | 10% noise |
|---|---|---|---|
| IGL (full CI) | $32.60 \pm 0.24$ | $33.75 \pm 0.29$ | $33.33 \pm 0.29$ |
| AI-IGL | $\mathbf{89.10 \pm 3.58}$ | $\mathbf{76.60 \pm 4.09}$ | $\mathbf{64.25 \pm 4.16}$ |

Table 2: Policy accuracy for the simulated BCI experiments with action-inclusive feedback. Average and standard error reported over 20 trials for each algorithm.

## 6 Discussion

We have presented a new approach to solving Interaction-Grounded Learning, in which an agent learns to interact with the environment in the absence of any explicit reward signals. Compared to a prior solution [Xie et al., 2021b], the proposed AI-IGL approach removes the assumption of conditional independence of actions and the feedback vector by treating the latent reward as a latent state. It thereby provably solves IGL for action-inclusive feedback vectors.

By viewing the feedback as containing an action-(latent) reward pair which is an unobserved latent space, we propose latent reward learning using a contrastive learning approach. This solution concept naturally connects to latent state discovery in rich-observation reinforcement learning [e.g., Dann et al., 2018; Du et al., 2019; Misra et al., 2020]. On the other hand, different from rich-observation RL, the problem of IGL also contains a unique challenge in identifying the semantic meaning of the decoded class, which is addressed by a symmetry-breaking procedure. In this work, we focus on binary latent rewards, for which symmetry breaking is possible using one policy (per action). Breaking symmetry in more general latent reward spaces is a topic for future work. A possible negative societal impact of this work can be performance instability, especially with inappropriate use of the techniques in risk-sensitive applications.

Barring intentional misuse, we envision several potential benefits of the proposed approach. The proposed algorithm broadens the scope of IGL's feasibility for real-world applications. Imagine an agent being trained to interpret the brain signals of a user to control a prosthetic arm. The brain's response (feedback vector) to an action is a neural signal that may contain information about the action itself. This is so prevalent in neuroimaging that fMRI studies routinely use specialized techniques to orthogonalize different information (e.g. action and reward) within the same signal [Momennejad and Haynes, 2012, 2013; Belghazi et al., 2018; Shah and Peters, 2020]. Another example is a continually self-calibrating eye tracker, used by people with motor disabilities such as ALS [Hansen et al., 2004; Liu et al., 2010; Mott et al., 2017; Gao et al., 2021]. A learning agent adapting to the ability of such users can encounter feedback directly influenced by the calibration correction action. The proposed approach is a stepping stone on the path to solving IGL for complex interactive settings, overcoming the need for explicit rewards as well as explicit separation of action and feedback information.

## Acknowledgments

The authors would like to thank Mark Rucker for sharing methods and code for computing and comparing diagnostic features of OpenML datasets. NJ acknowledges funding support from ARL Cooperative Agreement W911NF-17-2-0196, NSF IIS-2112471, NSF CAREER award, and Adobe Data Science Research Award.

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
