# Appendix

## A   Detailed Proofs

***Proof of Proposition 1.***   Over this proof, we follow the definition of Eq.(3) for each $(f, \psi) \in \mathcal{F} \times \Psi$. For any $\pi$, let $d_a^\pi := \sum_x d_0(x)\pi(a|x)$ and $\rho_a^\pi := 1/d_a^\pi \sum_x d_0(x)\pi(a|x) \Pr(r = 1|x, a)$, then

$$
\begin{aligned}
\mathbb{E}_{\mu_a}\left[f(x)\psi(y)\right] &= \frac{1}{d_a^\mu}\sum_{x,y} d_0(x)\mu(a|x)f(x)\Pr(y|a, r = 0)\psi(y)\Pr(r = 0|x, a) \\
&\quad + \frac{1}{d_a^\mu}\sum_{x,y} d_0(x)\mu(a|x)f(x)\Pr(y|a, r = 1)\psi(y)\Pr(r = 1|x, a) \\
&\hspace{6cm} \text{(by } x \perp\!\!\!\perp y|r, a) \\
&\overset{(a)}{=} (1 - \rho_a^\mu)f_{a,0}\psi_{a,0} + \rho_a^\mu f_{a,1}\psi_{a,1},
\end{aligned}
$$

and

$$
\begin{aligned}
\mathbb{E}_{\mu_a}\left[f(x)\right]\mathbb{E}_{\mu_a}\left[\psi(y)\right] &= \left(\frac{1}{d_a^\mu}\sum_x d_0(x)\mu(a|x)f(x)\right)\left(\frac{1}{d_a^\mu}\sum_{x,y} d_0(x)\mu(a|x)\Pr(y|a)\psi(y)\right) \\
&\overset{(b)}{=} \left((1 - \rho_a^\mu)f_{a,0} + \rho_a^\mu f_{a,1}\right)\left((1 - \rho_a^\mu)\psi_{a,0} + \rho_a^\mu\psi_{a,1}\right) \\
&= (1 - \rho_a^\mu)^2 f_{a,0}\psi_{a,0} + (\rho_a^\mu)^2 f_{a,1}\psi_{a,1} + (1 - \rho_a^\mu)\rho_a^\mu(f_{a,0}\psi_{a,1} + f_{a,1}\psi_{a,0}).
\end{aligned}
$$

To see why (a) and (b) hold, we use the following argument. By definition of $f_{a,r}$ and $\psi_{a,r}$, we have

$$
\begin{aligned}
f_{a,0} &= \sum_x \Pr(x|a, r = 0)f_a(x, a) \\
&= \sum_x \frac{\Pr(x, a, r = 0)f_a(x, a)}{\Pr(a, r = 0)} \\
&= \frac{\sum_x d_0(x)\mu(a|x)f_a(x, a)\Pr(r = 0|x, a)}{\sum_x d_0(x)\mu(a|x)\Pr(r = 0|x, a)}. \\
\Longrightarrow d_a^\mu(1 - \rho_a^\mu)f_{a,0} &= \sum_x d_0(x)\mu(a|x)f_a(x, a)\Pr(r = 0|x, a).
\end{aligned}
$$

Then, we have

$$
\begin{aligned}
&\frac{1}{d_a^\mu}\sum_{x,y} d_0(x)\mu(a|x)f_a(x, a)\Pr(y|a, r = 0)\psi_a(y, a)\Pr(r = 0|x, a) \\
&= \frac{1}{d_a^\mu}\sum_x d_0(x)\mu(a|x)f_a(x, a)\Pr(r = 0|x, a)\sum_y \Pr(y|a, r = 0)\psi_a(y, a) \\
&= \frac{\psi_{a,0}}{d_a^\mu}\sum_x d_0(x)\mu(a|x)f_a(x, a)\Pr(r = 0|x, a) \\
&= (1 - \rho_a^\mu)f_{a,0}\psi_{a,0}.
\end{aligned}
$$

Similar procedure also induces the remaining terms of (a) and (b).

Therefore, combining the two equalities above, we obtain

$$
\begin{aligned}
&\mathbb{E}_{\mu_a}\left[f(x)\psi(y)\right] - \mathbb{E}_{\mu_a}\left[f(x)\right]\mathbb{E}_{\mu_a}\left[\psi(y)\right] \\
&= (1 - \rho_a^\mu)\rho_a^\mu(f_{a,0}\psi_{a,0} + f_{a,1}\psi_{a,1}) - (1 - \rho_a^\mu)\rho_a^\mu(f_{a,0}\psi_{a,1} + f_{a,1}\psi_{a,0}) \\
&= (1 - \rho_a^\mu)\rho_a^\mu(f_{a,0}\psi_{a,0} + f_{a,1}\psi_{a,1} - f_{a,0}\psi_{a,1} - f_{a,1}\psi_{a,0}) \\
&= (1 - \rho_a^\mu)\rho_a^\mu(f_{a,1} - f_{a,0})(\psi_{a,1} - \psi_{a,0}). \hspace{3cm} (6)
\end{aligned}
$$

This completes the proof. $\qquad\square$

**Lower bound of $\Delta_{\mathcal{F}}$ when $\mu = \pi_{\mathsf{unif}}$.** For any $f \in \mathcal{F}$, let $V_f(\pi) := \sum_x \mathbb{E}_{d_0 \times \pi}[f(x,a)] \in [0,1]$. Then,

$$
\mathbb{E}_\mu [f(x,\bar{a})|\bar{a}, r = 1] - \mathbb{E}_\mu [f(x,\bar{a})|\bar{a}, r = 0]
$$

$$
= \frac{\sum_x \Pr(x, \bar{a}, r = 1) f(x, \bar{a})}{\sum_x \Pr(x, \bar{a}, r = 1)} - \frac{\sum_x \Pr(x, \bar{a}, r = 0) f(x, \bar{a})}{\sum_x \Pr(x, \bar{a}, r = 0)}
$$

$$
= \frac{\sum_x d_0(x)\mu(\bar{a}|x) R(x, \bar{a}) f(x, \bar{a})}{\sum_x d_0(x)\mu(\bar{a}|x) R(x, \bar{a})} - \frac{\sum_x d_0(x)\mu(\bar{a}|x) f(x, \bar{a})(1 - R(x, \bar{a}))}{\sum_x d_0(x)\mu(\bar{a}|x)(1 - R(x, \bar{a}))}
$$

$$
= \frac{\sum_x d_0(x) R(x, \bar{a}) f(x, \bar{a})}{\sum_x d_0(x) R(x, \bar{a})} - \frac{\sum_x d_0(x) f(x, \bar{a})(1 - R(x, \bar{a}))}{\sum_x d_0(x)(1 - R(x, \bar{a}))} \qquad (\mu = \pi_{\mathsf{unif}})
$$

$$
= \frac{\sum_x d_0(x) f(x, \bar{a}) R(x, \bar{a})}{V(\pi_{\bar{a}})} - \frac{\sum_x d_0(x) f(x, \bar{a})(1 - R(x, \bar{a}))}{1 - V(\pi_{\bar{a}})}
$$

$$
(\pi_{\bar{a}} \text{ denotes the constant policy with action } \bar{a})
$$

$$
= \frac{\sum_x d_0(x) f(x, \bar{a}) R(x, \bar{a}) - V_f(\pi_{\bar{a}}) V(\pi_{\bar{a}})}{V(\pi_{\bar{a}})}
$$

$$
- \frac{\sum_x d_0(x) f(x, \bar{a})(1 - R(x, \bar{a})) - V_f(\pi_{\bar{a}})(1 - V(\pi_{\bar{a}}))}{1 - V(\pi_{\bar{a}})}
$$

$$
= \frac{\mathrm{Cov}_{\pi_{\bar{a}}}(f, R)}{V(\pi_{\bar{a}})} - \frac{\mathrm{Cov}_{\pi_{\bar{a}}}(f, 1 - R)}{1 - V(\pi_{\bar{a}})}
$$

$$
\overset{(a)}{=} \frac{\mathrm{Cov}_{\pi_{\bar{a}}}(f, R)}{V(\pi_{\bar{a}})} + \frac{\mathrm{Cov}_{\pi_{\bar{a}}}(f, R)}{1 - V(\pi_{\bar{a}})}
$$

$$
= \frac{\mathrm{Cov}_{\pi_{\bar{a}}}(f, R)}{V(\pi_{\bar{a}})(1 - V(\pi_{\bar{a}}))}
$$

$$
\implies |\mathbb{E}_\mu [f(x, \bar{a})|\bar{a}, r = 1] - \mathbb{E}_\mu [f(x, \bar{a})|\bar{a}, r = 0]| \geq 4|\mathrm{Cov}_{\pi_{\bar{a}}}(f, R)|,
$$

where (a) follows from

$$
\mathrm{Cov}_{\pi_{\bar{a}}}(f, 1 - R) = \mathbb{E}_{\pi_{\bar{a}}}[(f(x, a) - V_f(\pi_{\bar{a}}))(1 - R(x, a) - 1 - V(\pi_{\bar{a}}))]
$$
$$
= -\mathbb{E}_{\pi_{\bar{a}}}[(f(x, a) - V_f(\pi_{\bar{a}}))(R(x, a) - V(\pi_{\bar{a}}))]
$$
$$
= -\mathrm{Cov}_{\pi_{\bar{a}}}(f, R).
$$

This completes the proof. $\qquad \square$

***Proof of Lemma 3.*** Let,

$$
(\widetilde{f}_{\bar{a}}, \widetilde{\psi}_{\bar{a}}) \leftarrow \underset{(f, \psi) \in \mathcal{F} \times \Psi}{\mathrm{argmax}} \; \mathcal{J}_{\bar{a}, \mu}(f, \psi).
$$

Over this proof, we define

$$
\Delta_{\mathcal{F}} = \max_{f \in \mathcal{F}} (f_{a,1} - f_{a,0}).
$$

Now, for $\mathcal{J}_{\bar{a}, \mathcal{D}}(f, \psi)$ with any $(f, \psi)$, we also have with probability at least $1 - \delta$

$$
|\mathcal{J}_{\bar{a}, \mathcal{D}}(f, \psi) - \mathcal{J}_{\bar{a}, \mu}(f, \psi)| \leq \varepsilon_{\mathsf{stat}, \bar{a}}.
$$

This means

$$
\mathcal{J}_{\bar{a}, \mathcal{D}}(\widetilde{f}_{\bar{a}}, \widetilde{\psi}_{\bar{a}}) \leq \mathcal{J}_{\bar{a}, \mathcal{D}}(f_{\bar{a}}, \psi_{\bar{a}})
$$
$$
\implies \mathcal{J}_{\bar{a}, \mu}(\widetilde{f}_{\bar{a}}, \widetilde{\psi}_{\bar{a}}) - 2\varepsilon_{\mathsf{stat}, \bar{a}} \leq \mathcal{J}_{\bar{a}, \mu}(f_{\bar{a}}, \psi_{\bar{a}})
$$
$$
\implies \underbrace{(1 - \rho_a^\mu)\rho_a^\mu}_{=K-1/K^2, \text{ as } \mu = \pi_{\mathsf{unif}}} \cdot (\widetilde{f}_{\bar{a},1} - \widetilde{f}_{\bar{a},0})(\widetilde{\psi}_{\bar{a},1} - \widetilde{\psi}_{\bar{a},0}) - 2\varepsilon_{\mathsf{stat}, \bar{a}} \leq (1 - \rho_a^\mu)\rho_a^\mu (f_{\bar{a},1} - f_{\bar{a},0})(\psi_{\bar{a},1} - \psi_{\bar{a},0}).
$$

$$
(\text{by Eq.}(6))
$$

$$
\implies (f_{\bar{a},1} - f_{\bar{a},0})(\psi_{\bar{a},1} - \psi_{\bar{a},0}) \geq (\widetilde{f}_{\bar{a},1} - \widetilde{f}_{\bar{a},0})(\widetilde{\psi}_{\bar{a},1} - \widetilde{\psi}_{\bar{a},0}) - \frac{2K^2}{(K - 1)}\varepsilon_{\mathsf{stat}, \bar{a}}.
$$

This implies

$$|f_{\bar{a},1} - f_{\bar{a},0}| \geq \Delta_{\mathcal{F}} - \frac{2K^2}{(K-1)}\varepsilon_{\mathsf{stat},\bar{a}}$$

$$|\psi_{\bar{a},1} - \psi_{\bar{a},0}| \geq 1 - \frac{2K^2}{\Delta_{\mathcal{F}}(K-1)}\varepsilon_{\mathsf{stat},\bar{a}}$$

$$\implies |\psi_{\bar{a},1}^{\star} - \max\{\psi_{\bar{a},1}, \psi_{\bar{a},0}\}|, |\min\{\psi_{\bar{a},1}, \psi_{\bar{a},0}\} - \psi_{\bar{a},0}^{\star}| \leq \frac{2K^2}{\Delta_{\mathcal{F}}(K-1)}\varepsilon_{\mathsf{stat},\bar{a}}. \tag{7}$$

This completes the proof. $\qquad\square$

***Proof of Lemma 4.*** We now provide the proof for any fixed $\bar{a} \in \mathcal{A}$. We define $d_{\bar{a}}^{\pi} := \sum_x d_0(x)\pi(\bar{a}|x)$, and $R_{\psi}(x,a) = \mathbb{E}[\psi(y,a)|x,a]$ as the decoded reward by $\psi$ for all $(x,a,\psi) \in \mathcal{X} \times \mathcal{A} \times \Psi$. Also, let $\rho_{\bar{a}}^{\pi_{\mathsf{base}}^{\bar{a}}}, \widehat{\rho}_{\bar{a}}^{\pi_{\mathsf{base}}^{\bar{a}}}, \varrho_{\bar{a}}^{\pi_{\mathsf{base}}^{\bar{a}}}, \widehat{\varrho}_{\bar{a}}^{\pi_{\mathsf{base}}^{\bar{a}}}$ be,

$$\rho_{\bar{a}}^{\pi_{\mathsf{base}}^{\bar{a}}} = \frac{\overbrace{\sum_x d_0(x)\pi_{\mathsf{base}}^{\bar{a}}(\bar{a}|x)R(x,\bar{a})}^{=:(\mathrm{I}).\mathsf{pop}}}{\underbrace{\sum_x d_0(x)\pi_{\mathsf{base}}^{\bar{a}}(\bar{a}|x)}_{=:(\mathrm{II}).\mathsf{pop}}}, \qquad \widehat{\rho}_{\bar{a}}^{\pi_{\mathsf{base}}^{\bar{a}}} = \frac{\overbrace{\sum_{(x,a,y)\sim\mathcal{D}}\pi_{\mathsf{base}}^{\bar{a}}(a|x)R(x,a)\mathbb{1}(a=\bar{a})}^{=:\sum_{(x,a,y)\sim\mathcal{D}}\mathbb{1}(a=\bar{a})\cdot(\mathrm{I}).\mathsf{emp}}}{\underbrace{\sum_{(x,a,y)\sim\mathcal{D}}\pi_{\mathsf{base}}^{\bar{a}}(a|x)\mathbb{1}(a=\bar{a})}_{=:\sum_{(x,a,y)\sim\mathcal{D}}\mathbb{1}(a=\bar{a})\cdot(\mathrm{II}).\mathsf{emp}}} \tag{8}$$

$$\varrho_{\bar{a}}^{\pi_{\mathsf{base}}} = \frac{\overbrace{\sum_x d_0(x)\pi_{\mathsf{base}}^{\bar{a}}(\bar{a}|x)R_{\psi_{\bar{a}}}(x,\bar{a})}^{:=(\mathrm{III}).\mathsf{pop}}}{\sum_x d_0(x)\pi_{\mathsf{base}}^{\bar{a}}(\bar{a}|x)}, \qquad \widehat{\varrho}_{\bar{a}}^{\pi_{\mathsf{base}}^{\bar{a}}} = \frac{\overbrace{\sum_{(x,a,y)\sim\mathcal{D}}\pi_{\mathsf{base}}^{\bar{a}}(a|x)\psi_a(y,a)\mathbb{1}(a=\bar{a})}^{=:\sum_{(x,a,y)\sim\mathcal{D}}\mathbb{1}(a=\bar{a})\cdot(\mathrm{III}).\mathsf{emp}}}{\sum_{(x,a,y)\sim\mathcal{D}}\pi_{\mathsf{base}}^{\bar{a}}(a|x)\mathbb{1}(a=\bar{a})} \tag{9}$$

Over this section, we define $\varepsilon_{\mathsf{stat},\bar{a}}$ as,

$$\varepsilon_{\mathsf{stat},\bar{a}} := \mathcal{O}\left(\sqrt{\frac{Kd_{\mathcal{F},\Psi}}{|\mathcal{D}|}}\right) \geq \mathcal{O}\left(\sqrt{\frac{d_{\mathcal{F},\Psi}}{\sum_{(x,a,y)\sim\mathcal{D}}\mathbb{1}(a=\bar{a})}}\right).$$
$$\text{(by [Xie et al., 2021a, Lemma A.1])}$$

Then, we know with probability at least $1 - \delta$,

$$|(\mathrm{I}).\mathsf{pop} - (\mathrm{I}).\mathsf{emp}|, |(\mathrm{II}).\mathsf{pop} - (\mathrm{II}).\mathsf{emp}|, |(\mathrm{III}).\mathsf{pop} - (\mathrm{III}).\mathsf{emp}| \leq \varepsilon_{\mathsf{stat},\bar{a}} \tag{10}$$

by a standard concentration argument.

We now show when line 5-6 in Algorithm 1 correctly break the symmetry. By Assumption 4, we have $\rho_{\bar{a}}^{\pi_{\mathsf{base}}^{\bar{a}}} \leq \frac{1}{2} - \eta$. In addition, by Eq.(10),

$$\left|\widehat{\rho}_{\bar{a}}^{\pi_{\mathsf{base}}^{\bar{a}}} - \rho_{\bar{a}}^{\pi_{\mathsf{base}}^{\bar{a}}}\right|, \left|\widehat{\varrho}_{\bar{a}}^{\pi_{\mathsf{base}}^{\bar{a}}} - \varrho_{\bar{a}}^{\pi_{\mathsf{base}}^{\bar{a}}}\right| \geq 2\varepsilon_{\mathsf{stat},\bar{a}} + \frac{2\varepsilon_{\mathsf{stat},\bar{a}}^2}{c_m - \varepsilon_{\mathsf{stat},\bar{a}}}. \tag{11}$$

Recall the definition of $\rho_{\bar{a}}^{\pi_{\mathsf{base}}^{\bar{a}}}$ and $\varrho_{\bar{a}}^{\pi_{\mathsf{base}}^{\bar{a}}}$ (Eq.(8) and Eq.(9)), we have

$$\rho_{\bar{a}}^{\pi_{\mathsf{base}}^{\bar{a}}} - \varrho_{\bar{a}}^{\pi_{\mathsf{base}}^{\bar{a}}} = \frac{\sum_x d_0(x)\pi_{\mathsf{base}}^{\bar{a}}(\bar{a}|x)\mathbb{E}[\psi^{\star}(y,\bar{a}) - \psi_{\bar{a}}(y,\bar{a})|x,\bar{a}]}{\sum_x d_0(x)\pi_{\mathsf{base}}^{\bar{a}}(\bar{a}|x)R_{\psi_{\bar{a}}}(x,\bar{a})}.$$

According to Eq.(7), if $\psi_{\bar{a},1} > \psi_{\bar{a},0}$, we have

$$\varrho_{\bar{a}}^{\pi_{\mathsf{base}}^{\bar{a}}} \leq \rho_{\bar{a}}^{\pi_{\mathsf{base}}^{\bar{a}}} + \frac{2K^2}{(K-1)}\varepsilon_{\mathsf{stat},\bar{a}}$$

$$\implies \widehat{\varrho}_{\bar{a}}^{\pi_{\mathsf{base}}^{\bar{a}}} \leq \rho_{\bar{a}}^{\pi_{\mathsf{base}}^{\bar{a}}} + 2\varepsilon_{\mathsf{stat},\bar{a}} + \frac{2\varepsilon_{\mathsf{stat},\bar{a}}^2}{c_m - \varepsilon_{\mathsf{stat},\bar{a}}}, \qquad \text{(by Eq.(11))}$$

otherwise,

$$\rho_{\bar{a}}^{\pi_{\mathsf{base}}^{\bar{a}}} - \varrho_{\bar{a}}^{\pi_{\mathsf{base}}^{\bar{a}}} = 1 - \frac{\sum_x d_0(x)\pi_{\mathsf{base}}(\bar{a}|x)\mathbb{E}[1 - \psi_{\bar{a}}(y,\bar{a}) - \psi^\star(y,\bar{a})|x,\bar{a}]}{\sum_x d_0(x)\pi_{\mathsf{base}}(\bar{a}|x)R_{\psi_{\bar{a}}}(x,\bar{a})}$$

$$\implies \varrho_{\bar{a}}^{\pi_{\mathsf{base}}^{\bar{a}}} \geq 1 - \rho_{\bar{a}}^{\pi_{\mathsf{base}}^{\bar{a}}} - \frac{2K^2}{\Delta_{\mathcal{F}}(K-1)}\varepsilon_{\mathsf{stat},\bar{a}}$$

$$\implies \widehat{\varrho}_{\bar{a}}^{\pi_{\mathsf{base}}^{\bar{a}}} \geq 1 - \rho_{\bar{a}}^{\pi_{\mathsf{base}}^{\bar{a}}} - \frac{2K^2}{\Delta_{\mathcal{F}}(K-1)}\varepsilon_{\mathsf{stat},\bar{a}} - 2\varepsilon_{\mathsf{stat},\bar{a}} - \frac{2\varepsilon_{\mathsf{stat},\bar{a}}^2}{c_m - \varepsilon_{\mathsf{stat},\bar{a}}}. \qquad \text{(by Eq.(11))}$$

To guarantee the correctness of the symmetry breaking step (line 5-6 in Algorithm 1), we need

1. If $f_{\bar{a},1} > f_{\bar{a},0}$, $\widehat{\varrho}_{\bar{a}}^{\pi_{\mathsf{base}}^{\bar{a}}} \leq \frac{1}{2}$,

2. Otherwise, $\widehat{\varrho}_{\bar{a}}^{\pi_{\mathsf{base}}^{\bar{a}}} > \frac{1}{2}$.

That requires,

$$\frac{2K^2}{\Delta_{\mathcal{F}}(K-1)}\varepsilon_{\mathsf{stat},\bar{a}} + 2\varepsilon_{\mathsf{stat},\bar{a}} + \frac{2\varepsilon_{\mathsf{stat},\bar{a}}^2}{c_m - \varepsilon_{\mathsf{stat},\bar{a}}} \leq \eta.$$

This can be induced by

$$\varepsilon_{\mathsf{stat},\bar{a}} \leq \mathcal{O}\left(\min\{\eta\Delta_{\mathcal{F}}/K, c_m\}\right) \quad \left(\impliedby \quad |\mathcal{D}| \geq \mathcal{O}\left(\frac{K^3 d_{\mathcal{F},\Psi}}{(\min\{\eta\Delta_{\mathcal{F}}, Kc_m\})^2}\right)\right). \qquad \square$$

***Proof of Theorem 2.*** Combining Lemma 3 and Lemma 4, we know for any $\bar{a} \in \mathcal{A}$

$$|\psi_{\bar{a},1}^\star - \psi_{\bar{a},1}|, |\psi_{\bar{a},0} - \psi_{\bar{a},0}^\star| \leq \frac{2K^2}{\Delta_{\mathcal{F}}(K-1)}\varepsilon_{\mathsf{stat},\bar{a}}.$$

Then,

$$\begin{aligned}
&V(\pi^\star) - V(\widehat{\pi})\\
&= \mathbb{E}_{d_0 \times \pi^\star}[r] - \mathbb{E}_{d_0 \times \widehat{\pi}}[r]\\
&= \mathbb{E}_{(x,a,y)\sim d_0 \times \pi^\star}[\psi^\star(y,a)] - \mathbb{E}_{(x,a,y)\sim d_0 \times \widehat{\pi}}[\psi^\star(y,a)] && \text{(by Assumption 3)}\\
&= \mathbb{E}_{(x,a,y)\sim d_0 \times \pi^\star}[\psi_a'(y,a)] - \mathbb{E}_{(x,a,y)\sim d_0 \times \widehat{\pi}}[\psi_a'(y,a)]\\
&\quad + \left|\mathbb{E}_{(x,a,y)\sim d_0 \times \pi^\star}[\psi^\star(y,a) - \psi_a'(y,a)]\right| + \left|\mathbb{E}_{(x,a,y)\sim d_0 \times \widehat{\pi}}[\psi^\star(y,a) - \psi_a'(y,a)]\right|\\
&\leq \mathbb{E}_{(x,a,y)\sim d_0 \times \pi^\star}[\psi_a'(y,a)] - \mathbb{E}_{(x,a,y)\sim d_0 \times \widehat{\pi}}[\psi_a'(y,a)]\\
&\quad + (V(\pi^\star) + V(\widehat{\pi}))\max_{a\in\mathcal{A}}|\psi_{\bar{a},1}^\star - \psi_{\bar{a},1}| + (2 - V(\pi^\star) - V(\widehat{\pi}))\max_{a\in\mathcal{A}}|\psi_{\bar{a},0} - \psi_{\bar{a},1}^\star|\\
&\leq \mathbb{E}_{(x,a,y)\sim d_0 \times \pi^\star}[\psi_a'(y,a)] - \mathbb{E}_{(x,a,y)\sim d_0 \times \pi^\star}[\psi_a'(y,a)] + \frac{4K^2}{\Delta_{\mathcal{F}}(K-1)}\varepsilon_{\mathsf{stat},\bar{a}}\\
&\leq \varepsilon_{\mathsf{CB}} + \frac{4K^3}{\Delta_{\mathcal{F}}(K-1)}\varepsilon_{\mathsf{stat},\bar{a}}. && \text{(by the property of CB oracle in Definition 1)}
\end{aligned}$$

This completes the proof. $\qquad \square$

## B  Experiments on the MNIST dataset

The environment for this experiment is generated from the supervised classification MNIST dataset [LeCun et al., 1998] which is licensed under Attribution-Share Alike 3.0 license[4]. At each time step, the context $x_t$ is generated uniformly at random. Then the learner selects an action $a_t \in \{0, \ldots, 9\}$ as the predicted label of $x_t$. The binary reward $r$ is the correctness of the prediction label $a_t$. The high-dimensional feedback vector $y_t$ is an image of the digit $(a_t + 6r - 3) \mod 10$. An example is shown in Figure 3. Our results are averaged over 20 trials.

---

[4]https://creativecommons.org/licenses/by-sa/3.0/

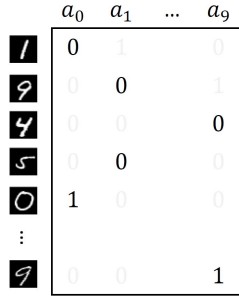
(a) Contextual Bandits (CB)

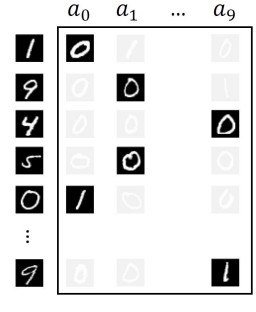
(b) IGL (full CI) [Xie et al., 2021b]

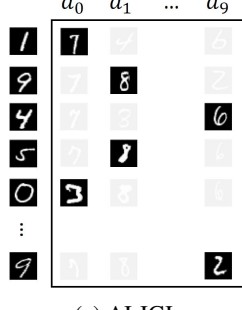
(c) AI-IGL

Figure 3: Different learning approaches based on the MNIST dataset. The gray number/image denotes the unobserved reward/feedback vector. **Figure 3(a):** In the contextual bandits setting, the exact reward information on the selected action can be observed. **Figure 3(b):** In IGL (full CI), the feedback vector is generated only based on the latent reward. **Figure 3(c):** In AI-IGL, the feedback vector can be generated based on both latent reward and selected action.

| Algorithm | Policy accuracy for action-inclusive feedback (%) | Policy accuracy for action-exclusive feedback (%) |
|---|---|---|
| CB | **87.64 ± 0.25** | **87.64 ± 0.25** |
| IGL (full CI) | 9.18 ± 2.10 | 86.13 ± 0.52 |
| AI-IGL | **83.63 ± 1.25** | **86.14 ± 0.94** |

Table 3: Results in the MNIST environment with high-dimensional action-inclusive and action-exclusive feedback. Average and standard error reported over 20 trials for each algorithm.

To highlight that the proposed algorithm still operates well under conditions where the feedback vector does not include action information, we also perform experiments under the setting introduced by Xie et al. [2021b] on the MNIST environment. This setting is similar to the one described in Section B except the feedback vector is the image of the digit $r$ instead of $(a_t + 6r - 3) \mod 10$. We find that under this action-exclusive feedback setting, the new proposed algorithm AI-IGL works as well as the IGL (full CI). This signifies that our algorithm, which incorporates the presence of action information in the feedback vector, does not hurt performance in cases when the action information is missing from the feedback vector.

## C  Additional Experimental Details

This section provides additional details on our implementation. The experiments on the MNIST dataset were conducted using a Google Colab, which was based on Intel Xeon CPU (2.30GHz) and 12 GB memory. Large scale experiments on the OpenML datasets were conducted on CPU instances of an internal cluster. No GPU was used. The prototype codes were built over Python, PyTorch[5], and Vowpal Wabbit[6]. With the single process in the setup above, a single trial of each experiment took less than 30 minutes to finish. Each experiment runs for 10 epochs over the dataset it is being evaluated on. The data is shuffled only once before the training begins so each algorithms views the data in the same order in each epoch.

The BCI experiment is established based on the Brain Imaging Analysis Kit (brainiak)[7], which provides the realistic simulation of functional Magnetic Resonance Imaging (fMRI) based on the real human data [Bejjanki et al., 2017]. Our detailed setup, including the choice of the most of experiment parameters, follows from the demo provided in brainiak[8].

---

[5]https://pytorch.org/

[6]https://vowpalwabbit.org/

[7]https://github.com/brainiak/brainiak

[8]https://github.com/brainiak/brainiak/blob/master/examples/utils/fmrisim_multivariate_example.ipynb

## C.1 Results for Individual OpenML datasets

For each individual OpenML dataset, we report the mean performance and standard errors for the best contact action policy, contextual bandit (CB), IGL assuming conditional independence of feedback on context and action given the latent reward (full CI) [Xie et al., 2021b], and IGL with conditional independence of feedback on context given reward (AI-IGL) in Table 4. The best constant action policy is computed based on the highest percentage of samples in the entire dataset with the same action. We do not report the standard error for it.

Table 4: Average accuracy and standard errors for different algorithms on individual OpenML datasets.

| Dataset ID | N | K | Best Constant Action Policy (%) | CB (%) | IGL (full CI) (%) | AI-IGL (%) |
|---|---|---|---|---|---|---|
| 6 | 20000 | 26 | 4.07 | 53.90±0.53 | 3.54±0.18 | 11.40±1.45 |
| 7 | 226 | 24 | 25.22 | 34.35±2.70 | 5.65±1.69 | 8.26±2.26 |
| 9 | 205 | 6 | 32.68 | 42.38±2.56 | 16.90±2.98 | 20.95±3.25 |
| 11 | 625 | 3 | 46.08 | 85.79±1.07 | 32.38±4.33 | 25.00±6.60 |
| 12 | 2000 | 10 | 10.00 | 87.10±0.79 | 10.17±1.04 | 32.17±4.25 |
| 14 | 2000 | 10 | 10.00 | 67.75±0.77 | 10.98±0.68 | 20.47±3.15 |
| 16 | 2000 | 10 | 10.00 | 87.37±0.42 | 9.85±0.73 | 48.52±4.34 |
| 18 | 2000 | 10 | 10.00 | 59.13±0.94 | 9.25±0.71 | 26.48±3.48 |
| 20 | 2000 | 10 | 10.00 | 92.25±0.49 | 9.82±0.75 | 43.35±4.35 |
| 22 | 2000 | 10 | 10.00 | 71.55±0.88 | 9.88±0.81 | 23.85±3.64 |
| 23 | 1473 | 3 | 42.70 | 48.85±0.92 | 38.61±1.72 | 31.79±1.66 |
| 26 | 12960 | 5 | 33.33 | 89.86±0.23 | 19.66±3.31 | 77.34±2.83 |
| 28 | 5620 | 10 | 10.18 | 93.99±0.20 | 10.30±0.75 | 66.49±4.47 |
| 32 | 10992 | 10 | 10.41 | 88.10±0.22 | 9.62±0.59 | 74.47±3.01 |
| 35 | 366 | 6 | 30.60 | 95.00±0.82 | 12.16±1.62 | 46.62±6.06 |
| 36 | 2310 | 7 | 14.29 | 87.27±0.64 | 13.35±1.00 | 63.35±4.70 |
| 39 | 336 | 8 | 42.56 | 71.76±2.20 | 11.47±2.69 | 23.53±4.86 |
| 41 | 214 | 6 | 35.51 | 49.55±2.87 | 14.55±2.96 | 22.05±3.68 |
| 42 | 683 | 19 | 13.47 | 60.22±1.89 | 4.57±0.70 | 20.36±2.46 |
| 48 | 151 | 3 | 34.44 | 42.19±3.03 | 27.19±2.09 | 31.25±2.30 |
| 54 | 846 | 4 | 25.77 | 50.12±1.56 | 23.88±0.91 | 28.94±2.19 |
| 60 | 5000 | 3 | 33.84 | 86.02±0.27 | 33.85±0.48 | 68.82±4.89 |
| 61 | 150 | 3 | 33.33 | 89.00±1.90 | 33.33±2.87 | 37.67±6.67 |
| 62 | 101 | 7 | 40.59 | 75.00±4.40 | 10.91±3.99 | 23.18±5.32 |
| 74 | 1000000 | 26 | 4.08 | 41.96±0.30 | 3.87±0.59 | 33.08±4.14 |
| 75 | 1000000 | 7 | 32.33 | 60.04±0.09 | 15.91±3.47 | 58.35±0.28 |
| 78 | 1000000 | 10 | 10.06 | 83.61±0.05 | 10.07±0.12 | 80.28±0.07 |
| 115 | 1000000 | 10 | 10.04 | 93.85±0.03 | 9.81±0.99 | 92.96±0.05 |
| 116 | 1000000 | 6 | 42.27 | 67.41±0.06 | 16.20±1.90 | 66.11±0.06 |
| 117 | 1000000 | 6 | 42.27 | 66.79±0.08 | 19.84±5.02 | 65.11±0.10 |
| 118 | 1000000 | 10 | 10.04 | 76.10±0.07 | 9.58±0.82 | 72.91±0.02 |
| 119 | 55296 | 3 | 42.78 | 48.76±0.17 | 33.59±1.75 | 40.31±1.91 |
| 123 | 1000000 | 10 | 10.17 | 93.02±0.04 | 11.90±1.77 | 90.73±0.02 |
| 127 | 1000000 | 10 | 10.46 | 80.56±0.06 | 9.48±2.12 | 75.76±0.14 |
| 129 | 1000000 | 6 | 30.46 | 97.47±0.01 | 11.11±2.06 | 96.54±0.01 |
| 130 | 1000000 | 7 | 14.36 | 80.83±0.08 | 14.63±1.01 | 75.56±0.27 |
| 133 | 137781 | 7 | 35.04 | 59.08±0.14 | 15.86±2.34 | 47.74±3.06 |
| 134 | 1000000 | 19 | 13.33 | 88.33±0.05 | 5.64±1.05 | 83.17±0.04 |
| 141 | 1000000 | 4 | 25.81 | 58.92±0.03 | 22.28±1.56 | 58.59±0.17 |
| 147 | 1000000 | 3 | 33.78 | 83.32±0.10 | 33.22±0.14 | 82.46±0.07 |
| 148 | 1000000 | 7 | 39.62 | 93.46±0.05 | 16.52±2.19 | 92.25±0.01 |
| 149 | 1455525 | 10 | 44.97 | 61.58±0.07 | 4.11±1.93 | 46.16±6.88 |
| 150 | 581012 | 7 | 48.76 | 71.13±0.06 | 8.42±2.06 | 51.13±4.08 |
| 154 | 1000000 | 10 | 10.08 | 54.97±0.10 | 9.88±0.41 | 53.95±0.12 |
| 156 | 1000000 | 5 | 30.01 | 45.51±0.25 | 25.14±2.56 | 42.81±0.16 |
| 157 | 1000000 | 5 | 30.01 | 47.69±0.19 | 22.62±1.77 | 46.54±0.36 |
| 158 | 1000000 | 5 | 30.01 | 47.84±0.16 | 22.23±2.11 | 46.98±0.07 |
| 159 | 1000000 | 5 | 30.01 | 29.89±0.09 | 18.77±1.09 | 12.02±2.21 |
| 160 | 1000000 | 5 | 30.01 | 30.52±0.06 | 16.17±1.79 | 13.51±1.70 |
| 163 | 32 | 3 | 40.62 | 38.75±4.50 | 25.00±4.33 | 35.00±5.70 |
| 171 | 339 | 21 | 24.78 | 23.82±2.13 | 3.53±1.11 | 7.21±2.14 |
| 180 | 110393 | 7 | 46.82 | 64.34±0.11 | 10.34±1.86 | 40.44±3.94 |
| 181 | 1484 | 10 | 31.20 | 47.05±1.22 | 6.31±1.17 | 21.07±2.09 |
| 182 | 6430 | 6 | 23.81 | 82.20±0.36 | 18.99±1.40 | 76.87±0.88 |
| 183 | 4177 | 28 | 16.50 | 19.38±0.50 | 2.79±0.73 | 6.60±1.40 |
| 184 | 28056 | 18 | 16.23 | 23.94±0.32 | 5.71±0.63 | 6.95±0.83 |
| 187 | 178 | 3 | 39.89 | 93.06±1.41 | 30.56±2.65 | 37.50±6.65 |
| 188 | 736 | 5 | 29.08 | 49.59±1.06 | 18.85±1.46 | 21.96±3.81 |
| 247 | 1000000 | 26 | 4.08 | 42.91±0.28 | 3.08±0.48 | 26.60±1.03 |
| 248 | 1000000 | 7 | 32.36 | 54.00±0.06 | 8.53±3.26 | 51.59±0.33 |
| 250 | 1000000 | 10 | 10.05 | 82.52±0.05 | 9.81±0.33 | 78.84±0.10 |
| 252 | 1000000 | 10 | 10.04 | 96.43±0.01 | 9.63±0.59 | 95.62±0.04 |
| 253 | 1000000 | 6 | 42.31 | 65.27±0.05 | 12.10±1.77 | 50.18±11.17 |
| 254 | 1000000 | 10 | 10.03 | 74.63±0.04 | 11.11±1.23 | 72.07±0.09 |

Table 4: Average accuracy and standard errors for different algorithms on individual OpenML datasets.

| Dataset ID | N | K | Best Constant Action Policy (%) | CB (%) | IGL (full CI) (%) | AI-IGL (%) |
|---|---|---|---|---|---|---|
| 255 | 55296 | 3 | 42.62 | 52.14±0.17 | 32.04±2.04 | 41.94±1.93 |
| 261 | 1000000 | 10 | 10.45 | 77.09±0.07 | 9.23±1.58 | 73.85±0.14 |
| 263 | 1000000 | 6 | 30.46 | 97.50±0.01 | 17.02±2.75 | 96.54±0.01 |
| 265 | 137781 | 7 | 35.40 | 53.82±0.14 | 15.45±2.60 | 38.90±2.25 |
| 268 | 1000000 | 4 | 25.75 | 57.85±0.11 | 24.47±0.60 | 57.02±0.16 |
| 271 | 1000000 | 3 | 33.84 | 88.70±0.04 | 33.14±0.07 | 87.99±0.12 |
| 272 | 1000000 | 7 | 39.65 | 93.41±0.05 | 17.01±3.15 | 91.99±0.05 |
| 285 | 194 | 8 | 30.93 | 31.00±2.36 | 12.25±2.25 | 19.75±2.68 |
| 300 | 7797 | 26 | 3.85 | 80.51±0.46 | 3.44±0.27 | 30.18±2.08 |
| 307 | 990 | 11 | 9.09 | 32.63±1.06 | 10.20±1.05 | 11.67±1.24 |
| 313 | 531 | 48 | 10.36 | 8.52±1.08 | 1.20±0.38 | 2.31±0.63 |
| 327 | 105 | 6 | 41.90 | 51.82±2.41 | 20.45±3.33 | 20.91±4.12 |
| 328 | 105 | 6 | 41.90 | 50.45±3.77 | 13.64±3.55 | 25.45±3.61 |
| 329 | 160 | 3 | 40.62 | 47.50±2.00 | 29.06±2.91 | 32.50±3.94 |
| 338 | 155 | 4 | 31.61 | 41.88±2.03 | 20.62±2.30 | 21.25±2.44 |
| 339 | 36 | 3 | 33.33 | 78.75±4.06 | 30.00±3.35 | 31.25±4.96 |
| 340 | 52 | 3 | 44.23 | 49.17±5.33 | 31.67±2.86 | 35.00±5.26 |
| 342 | 52 | 3 | 46.15 | 69.17±3.78 | 25.00±4.93 | 37.50±5.12 |
| 372 | 10108 | 46 | 28.47 | 26.67±0.59 | 1.38±0.21 | 12.17±1.05 |
| 375 | 9961 | 9 | 16.20 | 90.36±0.25 | 10.85±0.89 | 72.55±3.19 |
| 377 | 600 | 6 | 16.67 | 64.58±2.41 | 12.42±1.62 | 21.75±3.10 |
| 382 | 7019 | 8 | 27.61 | 36.33±0.69 | 10.42±0.72 | 20.98±1.52 |
| 383 | 690 | 10 | 23.19 | 70.94±1.61 | 10.43±1.17 | 25.00±3.73 |
| 385 | 927 | 7 | 37.97 | 90.16±1.16 | 15.48±2.25 | 27.31±4.33 |
| 386 | 913 | 10 | 17.20 | 56.79±1.08 | 10.76±0.99 | 14.67±2.17 |
| 387 | 414 | 9 | 31.88 | 70.48±2.09 | 7.62±1.57 | 19.52±2.89 |
| 388 | 204 | 6 | 44.61 | 67.62±2.93 | 18.10±3.13 | 29.52±5.76 |
| 389 | 2463 | 17 | 20.54 | 65.22±0.75 | 6.09±0.90 | 24.64±2.33 |
| 390 | 9558 | 44 | 7.28 | 47.65±0.47 | 2.25±0.11 | 15.37±0.97 |
| 391 | 1504 | 13 | 40.43 | 61.95±1.10 | 4.64±0.97 | 19.74±3.08 |
| 392 | 1003 | 10 | 19.34 | 66.34±1.14 | 8.61±0.58 | 21.78±2.74 |
| 393 | 3075 | 6 | 29.43 | 83.07±0.49 | 15.83±1.36 | 37.74±4.94 |
| 394 | 918 | 10 | 16.23 | 61.74±1.13 | 9.89±0.89 | 14.78±2.23 |
| 395 | 1657 | 25 | 22.39 | 47.74±0.83 | 2.35±0.31 | 13.40±2.00 |
| 396 | 3204 | 6 | 29.43 | 81.59±0.49 | 15.67±1.49 | 35.42±4.69 |
| 397 | 313 | 8 | 29.71 | 61.72±1.70 | 14.69±1.98 | 25.94±3.34 |
| 398 | 1560 | 20 | 21.86 | 55.80±1.31 | 3.27±0.58 | 18.17±2.81 |
| 399 | 11162 | 10 | 14.52 | 65.26±0.30 | 10.44±0.39 | 33.83±2.58 |
| 400 | 878 | 10 | 27.68 | 79.43±1.16 | 7.78±1.03 | 23.41±4.39 |
| 401 | 1050 | 10 | 15.71 | 59.90±1.03 | 9.05±0.86 | 15.52±2.27 |
| 452 | 285 | 7 | 41.40 | 35.69±2.32 | 16.38±2.70 | 18.10±2.42 |
| 457 | 27 | 4 | 44.44 | 51.67±6.86 | 30.00±7.42 | 23.33±5.82 |
| 458 | 841 | 4 | 37.69 | 99.59±0.15 | 23.35±2.67 | 59.82±8.04 |
| 460 | 379 | 4 | 37.20 | 50.53±1.64 | 21.71±1.74 | 30.13±2.37 |
| 468 | 72 | 6 | 16.67 | 48.75±4.83 | 20.62±3.22 | 18.12±4.28 |
| 469 | 797 | 6 | 19.45 | 19.31±0.91 | 16.75±1.01 | 16.88±1.07 |
| 473 | 2796 | 6 | 24.32 | 63.55±1.21 | 15.73±1.30 | 24.27±3.77 |
| 475 | 400 | 4 | 25.00 | 33.25±1.67 | 24.38±1.73 | 26.38±2.12 |
| 554 | 70000 | 10 | 11.25 | 89.92±0.10 | 9.98±0.29 | 84.17±0.12 |
| 679 | 1024 | 4 | 39.45 | 40.24±1.66 | 27.67±2.36 | 23.98±3.03 |
| 685 | 130 | 5 | 20.00 | 33.85±3.00 | 20.38±2.68 | 20.38±3.14 |
| 694 | 310 | 9 | 13.23 | 28.23±1.89 | 8.71±1.71 | 10.81±2.25 |
| 952 | 214 | 6 | 35.51 | 50.91±2.66 | 16.14±3.36 | 20.91±3.19 |
| 1041 | 3468 | 10 | 11.04 | 82.03±0.63 | 9.60±0.57 | 43.08±3.18 |
| 1044 | 10936 | 3 | 38.97 | 50.41±0.37 | 32.98±1.29 | 43.37±1.46 |
| 1079 | 95 | 5 | 28.42 | 46.50±5.49 | 17.50±2.90 | 16.50±3.62 |
| 1080 | 113 | 5 | 45.13 | 32.50±2.88 | 15.00±3.36 | 17.50±2.42 |
| 1081 | 89 | 4 | 48.31 | 38.33±3.89 | 23.33±2.93 | 23.89±4.03 |
| 1083 | 214 | 7 | 32.24 | 28.41±3.20 | 14.32±2.52 | 19.32±3.08 |
| 1088 | 383 | 9 | 40.47 | 47.56±3.16 | 10.00±1.88 | 12.56±2.23 |
| 1102 | 96 | 9 | 47.92 | 60.50±3.35 | 17.50±4.29 | 20.00±4.06 |
| 1106 | 190 | 14 | 15.79 | 22.63±2.91 | 7.89±1.21 | 6.05±1.36 |
| 1109 | 96 | 11 | 23.96 | 43.50±4.08 | 11.00±1.99 | 19.00±4.06 |
| 1115 | 151 | 3 | 34.44 | 47.19±2.71 | 28.12±2.14 | 29.69±1.82 |
| 1177 | 1000000 | 22 | 24.07 | 49.84±0.16 | 6.52±2.14 | 41.14±0.99 |
| 1183 | 1000000 | 6 | 23.84 | 80.99±0.13 | 23.14±3.79 | 76.32±0.21 |
| 1185 | 1000000 | 3 | 40.11 | 92.69±0.04 | 33.58±2.51 | 92.60±0.02 |
| 1186 | 1000000 | 5 | 28.98 | 46.57±0.17 | 17.83±2.19 | 46.59±0.12 |
| 1209 | 1000000 | 11 | 9.14 | 30.53±0.22 | 9.14±0.55 | 27.91±1.06 |
| 1214 | 1000000 | 9 | 16.08 | 77.46±0.09 | 8.53±0.90 | 74.37±0.35 |
| 1233 | 945 | 7 | 14.81 | 35.58±1.48 | 14.42±0.65 | 16.11±1.30 |
| 1378 | 1000000 | 26 | 4.08 | 34.15±0.36 | 3.98±0.44 | 20.64±4.44 |
| 1379 | 1000000 | 26 | 4.08 | 20.26±0.57 | 3.64±0.19 | 5.09±0.38 |
| 1380 | 1000000 | 26 | 4.08 | 13.97±0.46 | 3.79±0.16 | 4.01±1.18 |
| 1381 | 1000000 | 26 | 4.08 | 39.52±0.34 | 3.85±0.40 | 26.77±6.91 |
| 1382 | 1000000 | 26 | 4.08 | 34.60±0.34 | 4.02±0.43 | 26.70±2.34 |

Table 4: Average accuracy and standard errors for different algorithms on individual OpenML datasets.

| Dataset ID | N | K | Best Constant Action Policy (%) | CB (%) | IGL (full CI) (%) | AI-IGL (%) |
|---|---|---|---|---|---|---|
| 1383 | 1000000 | 26 | 4.08 | 29.49±0.20 | 4.08±0.28 | 16.45±5.71 |
| 1384 | 1000000 | 26 | 4.08 | 41.35±0.20 | 3.86±0.14 | 22.29±6.30 |
| 1385 | 1000000 | 26 | 4.08 | 37.62±0.14 | 3.92±0.50 | 21.30±5.19 |
| 1386 | 1000000 | 26 | 4.08 | 33.98±0.28 | 4.43±0.22 | 30.04±0.80 |
| 1387 | 1000000 | 24 | 24.14 | 74.78±0.05 | 2.85±1.01 | 66.79±0.07 |
| 1388 | 1000000 | 24 | 24.14 | 55.58±0.05 | 4.55±0.95 | 49.54±0.88 |
| 1389 | 1000000 | 24 | 24.14 | 44.84±0.11 | 3.43±0.50 | 34.55±1.30 |
| 1390 | 1000000 | 24 | 24.14 | 77.74±0.11 | 1.94±0.42 | 71.71±0.14 |
| 1391 | 1000000 | 24 | 24.14 | 74.80±0.05 | 2.80±0.79 | 67.01±0.31 |
| 1392 | 1000000 | 24 | 24.14 | 69.23±0.07 | 6.82±2.16 | 62.90±0.54 |
| 1393 | 1000000 | 7 | 32.36 | 46.65±0.12 | 20.86±4.63 | 36.27±1.42 |
| 1394 | 1000000 | 7 | 32.36 | 35.02±0.05 | 15.67±3.38 | 22.66±2.35 |
| 1395 | 1000000 | 7 | 32.36 | 33.36±0.07 | 7.12±2.60 | 4.77±2.63 |
| 1396 | 1000000 | 7 | 32.36 | 55.51±0.20 | 17.11±2.66 | 53.13±0.18 |
| 1397 | 1000000 | 7 | 32.36 | 46.56±0.13 | 14.76±4.15 | 37.56±1.30 |
| 1398 | 1000000 | 7 | 32.36 | 37.47±0.10 | 8.33±2.70 | 21.45±1.93 |
| 1399 | 1000000 | 7 | 32.36 | 53.28±0.09 | 12.66±2.06 | 49.73±0.46 |
| 1400 | 549796 | 7 | 32.43 | 53.06±0.05 | 20.55±2.25 | 46.79±0.41 |
| 1401 | 1000000 | 7 | 32.36 | 46.72±0.11 | 11.69±2.61 | 37.31±0.35 |
| 1413 | 150 | 3 | 33.33 | 88.67±2.63 | 31.00±2.22 | 35.33±7.57 |
| 1457 | 1500 | 50 | 2.00 | 11.60±0.64 | 1.90±0.32 | 2.30±0.31 |
| 1459 | 10218 | 10 | 13.86 | 32.69±0.39 | 8.52±0.57 | 9.77±1.22 |
| 1465 | 106 | 6 | 20.75 | 24.55±3.02 | 13.64±1.76 | 14.55±2.60 |
| 1466 | 2126 | 10 | 27.23 | 99.77±0.08 | 7.89±1.31 | 62.93±5.20 |
| 1468 | 1080 | 9 | 11.11 | 87.41±0.84 | 10.32±0.85 | 41.94±3.52 |
| 1472 | 768 | 37 | 9.64 | 13.90±1.14 | 3.83±0.73 | 2.73±0.78 |
| 1475 | 6118 | 6 | 41.75 | 46.27±0.45 | 17.03±1.63 | 24.70±2.06 |
| 1476 | 13910 | 6 | 21.63 | 94.80±0.25 | 17.51±1.00 | 87.76±1.11 |
| 1477 | 13910 | 6 | 21.63 | 96.46±0.18 | 15.03±1.34 | 84.52±1.91 |
| 1478 | 10299 | 6 | 18.88 | 95.02±0.29 | 17.29±1.27 | 91.06±2.23 |
| 1481 | 28056 | 18 | 16.23 | 23.34±0.31 | 5.33±0.61 | 7.39±1.05 |
| 1482 | 340 | 30 | 4.71 | 10.59±1.05 | 3.24±0.65 | 3.24±0.62 |
| 1483 | 164860 | 11 | 33.05 | 38.96±0.18 | 7.54±1.30 | 17.07±1.56 |
| 1491 | 1600 | 100 | 1.00 | 6.81±0.50 | 0.88±0.14 | 1.66±0.24 |
| 1492 | 1600 | 100 | 1.00 | 2.47±0.30 | 0.75±0.18 | 1.06±0.24 |
| 1493 | 1599 | 100 | 1.00 | 6.31±0.36 | 0.88±0.17 | 1.28±0.24 |
| 1497 | 5456 | 4 | 40.41 | 66.40±0.52 | 24.53±3.06 | 44.01±4.83 |
| 1499 | 210 | 3 | 33.33 | 70.95±3.87 | 32.62±1.66 | 25.95±4.74 |
| 1500 | 210 | 3 | 33.33 | 75.71±2.84 | 32.86±1.90 | 27.86±4.63 |
| 1501 | 1593 | 10 | 10.17 | 80.28±0.62 | 9.12±0.63 | 30.31±3.21 |
| 1503 | 263256 | 10 | 10.06 | 10.13±0.03 | 10.03±0.05 | 10.03±0.05 |
| 1508 | 403 | 5 | 32.01 | 67.68±1.44 | 19.39±2.34 | 24.63±3.92 |
| 1509 | 149332 | 22 | 14.73 | 23.68±0.41 | 5.07±0.67 | 7.76±0.43 |
| 1512 | 200 | 5 | 28.00 | 25.00±1.66 | 20.75±2.07 | 21.00±1.82 |
| 1513 | 123 | 5 | 39.02 | 31.54±3.12 | 19.23±3.24 | 26.92±3.46 |
| 1514 | 360 | 10 | 10.00 | 65.56±2.37 | 8.33±1.08 | 24.72±3.01 |
| 1515 | 571 | 20 | 10.51 | 18.28±1.29 | 6.29±0.73 | 7.07±1.21 |
| 1516 | 88 | 4 | 38.64 | 58.89±4.24 | 21.11±3.13 | 16.11±3.47 |
| 1517 | 47 | 5 | 42.55 | 31.00±4.12 | 17.00±3.54 | 18.00±5.08 |
| 1518 | 47 | 4 | 42.55 | 39.00±4.79 | 13.00±2.56 | 21.00±4.58 |
| 1520 | 164 | 5 | 28.66 | 42.94±2.28 | 19.41±2.73 | 22.35±4.27 |
| 1523 | 215 | 3 | 46.05 | 82.95±1.84 | 32.95±2.18 | 37.50±7.14 |
| 1525 | 5456 | 4 | 40.41 | 74.57±0.56 | 23.72±3.21 | 39.73±4.93 |
| 1548 | 2500 | 3 | 46.92 | 47.08±0.86 | 35.46±3.66 | 36.28±1.41 |
| 1549 | 742 | 8 | 22.10 | 18.20±0.99 | 11.87±0.98 | 12.80±1.31 |
| 1551 | 400 | 8 | 27.75 | 16.50±1.64 | 14.12±1.72 | 12.50±1.26 |
| 1552 | 1100 | 5 | 27.73 | 29.23±0.89 | 19.86±1.63 | 21.86±1.69 |
| 1553 | 700 | 3 | 35.00 | 38.71±1.18 | 30.43±1.06 | 34.93±1.16 |
| 1554 | 500 | 5 | 38.40 | 32.20±1.31 | 20.40±2.49 | 21.00±2.17 |
| 1555 | 976 | 8 | 23.57 | 15.26±1.09 | 11.43±0.79 | 12.60±0.83 |
| 1568 | 12958 | 4 | 33.34 | 89.77±0.25 | 23.83±2.77 | 83.15±1.89 |
| 1596 | 580943 | 7 | 48.76 | 69.45±0.07 | 18.54±3.50 | 36.97±2.49 |
| 4552 | 5665 | 102 | 8.88 | 51.48±1.22 | 0.39±0.12 | 15.04±1.60 |
| 40927 | 60000 | 10 | 10.00 | 30.01±0.69 | 9.98±0.17 | 13.24±1.30 |
| 40966 | 1080 | 8 | 13.89 | 47.27±2.21 | 13.52±1.14 | 22.18±2.34 |
| 40971 | 1000 | 30 | 8.00 | 10.40±0.71 | 4.45±0.63 | 4.75±0.83 |
| 40979 | 2000 | 10 | 10.00 | 92.40±0.46 | 10.32±0.97 | 38.15±3.68 |
| 40982 | 1941 | 7 | 34.67 | 64.08±0.61 | 13.31±1.97 | 31.62±3.78 |
| 40984 | 2310 | 7 | 14.29 | 79.94±0.81 | 16.15±1.25 | 50.71±4.97 |
| 40985 | 45781 | 20 | 6.35 | 6.28±0.13 | 5.05±0.22 | 5.33±0.33 |
| 40996 | 70000 | 10 | 10.00 | 83.51±0.14 | 9.61±0.45 | 80.84±0.16 |
| 41002 | 5880 | 3 | 44.47 | 97.57±0.16 | 32.84±1.91 | 83.48±5.57 |
| 41003 | 5880 | 3 | 49.61 | 87.84±0.40 | 34.97±2.99 | 65.33±6.96 |
| 41004 | 4704 | 3 | 44.20 | 97.44±0.18 | 31.80±1.46 | 92.30±3.63 |
| 41039 | 131600 | 47 | 2.13 | 51.52±0.18 | 2.25±0.08 | 16.85±0.86 |
| 41081 | 44557 | 10 | 18.82 | 15.47±0.80 | 8.84±0.44 | 10.35±0.38 |

Table 4: Average accuracy and standard errors for different algorithms on individual OpenML datasets.

| Dataset ID | N | K | Best Constant Action Policy (%) | CB (%) | IGL (full CI) (%) | AI-IGL (%) |
|---|---|---|---|---|---|---|
| 41082 | 9298 | 10 | 16.7 | 91.12±0.20 | 11.21±0.82 | 77.66±2.29 |
| 41083 | 400 | 40 | 2.50 | 4.88±1.33 | 2.88±0.69 | 2.00±0.45 |
| 41084 | 575 | 20 | 8.35 | 29.91±1.41 | 4.14±0.59 | 6.29±0.59 |
| 41163 | 10000 | 5 | 20.49 | 87.03±0.45 | 19.44±0.55 | 66.98±4.69 |
| 41164 | 8237 | 7 | 23.39 | 54.04±0.37 | 14.51±0.52 | 43.50±2.91 |
| 41165 | 10000 | 10 | 10.43 | 27.56±0.45 | 10.08±0.32 | 10.49±0.42 |
| 41166 | 58310 | 10 | 21.96 | 53.11±0.44 | 11.71±1.12 | 45.36±1.32 |
| 41167 | 416188 | 355 | 0.59 | 28.28±0.26 | 0.27±0.02 | 2.06±0.17 |
| 41168 | 83733 | 4 | 46.01 | 62.84±0.15 | 24.93±2.63 | 48.94±3.76 |
| 41169 | 65196 | 100 | 6.14 | 19.49±0.24 | 0.65±0.05 | 7.34±0.61 |
| 41511 | 150 | 3 | 33.33 | 84.67±3.09 | 34.33±2.59 | 52.00±7.14 |
| 41568 | 150 | 3 | 33.33 | 85.00±2.98 | 32.33±1.90 | 44.67±7.59 |
| 41583 | 150 | 3 | 33.33 | 82.67±3.00 | 36.67±1.91 | 39.67±8.47 |
| 41919 | 527 | 4 | 39.47 | 50.57±1.34 | 23.40±1.99 | 31.51±4.68 |
| 41939 | 218 | 5 | 38.53 | 35.23±2.31 | 23.64±3.21 | 23.64±2.22 |
| 41950 | 150 | 3 | 33.33 | 80.00±2.67 | 32.33±1.84 | 25.67±6.62 |
| 41960 | 523590 | 144 | 25.08 | 38.46±0.79 | 0.56±0.14 | 15.89±2.20 |
| 41972 | 9144 | 8 | 44.29 | 63.16±1.64 | 12.44±3.11 | 26.07±4.36 |
| 41981 | 296 | 14 | 30.74 | 26.33±1.65 | 4.83±1.09 | 12.00±1.77 |
| 41982 | 70000 | 10 | 10.00 | 75.86±0.12 | 10.11±0.25 | 70.14±0.14 |
| 41986 | 51839 | 43 | 5.79 | 87.55±0.17 | 3.23±0.29 | 50.86±1.25 |
| 41988 | 51839 | 43 | 5.79 | 89.76±0.14 | 2.97±0.31 | 56.77±1.73 |
| 41989 | 51839 | 43 | 5.79 | 88.43±0.17 | 1.61±0.16 | 45.18±1.15 |
| 41990 | 51839 | 43 | 5.79 | 15.89±0.13 | 1.54±0.16 | 8.34±0.42 |
| 41991 | 270912 | 49 | 2.58 | 53.96±0.13 | 2.17±0.08 | 26.62±0.65 |
| 41997 | 150 | 3 | 33.33 | 89.00±2.07 | 30.00±1.73 | 42.33±8.44 |
| 42003 | 150 | 3 | 33.33 | 82.67±3.00 | 34.67±2.19 | 16.67±5.30 |
| 42011 | 150 | 3 | 33.33 | 81.67±2.90 | 33.67±3.18 | 27.33±7.87 |
| 42016 | 150 | 3 | 33.33 | 83.67±2.96 | 32.00±2.39 | 32.67±7.66 |
| 42021 | 150 | 3 | 33.33 | 82.00±3.17 | 36.00±3.14 | 37.00±6.77 |
| 42026 | 150 | 3 | 33.33 | 83.00±3.28 | 31.33±2.27 | 38.33±7.01 |
| 42031 | 150 | 3 | 33.33 | 84.67±2.83 | 33.67±2.69 | 27.33±6.53 |
| 42036 | 150 | 3 | 33.33 | 85.67±2.68 | 37.00±2.23 | 44.67±7.57 |
| 42041 | 150 | 3 | 33.33 | 83.33±2.81 | 29.33±2.08 | 41.33±8.06 |
| 42046 | 150 | 3 | 33.33 | 86.67±2.58 | 32.67±3.09 | 30.33±6.90 |
| 42051 | 150 | 3 | 33.33 | 82.67±3.03 | 30.00±2.33 | 29.33±7.19 |
| 42056 | 150 | 3 | 33.33 | 80.33±3.35 | 33.00±2.52 | 28.67±7.29 |
| 42066 | 150 | 3 | 33.33 | 82.00±3.50 | 31.67±2.20 | 40.00±5.85 |
| 42071 | 150 | 3 | 33.33 | 83.33±2.56 | 29.67±2.60 | 43.67±6.95 |
| 42098 | 150 | 3 | 33.33 | 84.00±2.92 | 29.67±2.33 | 37.00±7.31 |
| 42140 | 9927 | 10 | 19.10 | 13.87±0.77 | 9.21±0.80 | 10.69±0.54 |
| 42141 | 49644 | 10 | 19.10 | 15.84±0.86 | 11.59±0.89 | 11.03±0.54 |
| 42186 | 150 | 3 | 33.33 | 82.33±2.92 | 32.67±2.26 | 28.00±7.27 |
| 42261 | 150 | 3 | 33.33 | 81.67±3.30 | 33.33±2.94 | 24.00±5.88 |
| 42345 | 70340 | 3 | 48.88 | 61.16±0.14 | 22.11±4.71 | 45.74±1.31 |
| 42396 | 108000 | 1000 | 0.10 | 4.78±0.09 | 0.11±0.01 | 0.42±0.03 |
| 42468 | 830000 | 5 | 20.22 | 68.18±0.14 | 21.15±1.19 | 60.11±0.39 |
| 42532 | 2778 | 10 | 27.29 | 45.90±0.91 | 9.95±0.78 | 21.89±2.99 |
| 42544 | 265 | 8 | 17.74 | 42.04±1.68 | 14.81±1.76 | 15.00±2.31 |
| 42585 | 344 | 3 | 44.19 | 75.14±1.62 | 39.43±2.45 | 26.57±6.01 |
| 42700 | 150 | 3 | 33.33 | 87.33±2.70 | 33.67±1.53 | 22.00±6.15 |
| 42718 | 1000000 | 4 | 25.75 | 57.78±0.10 | 24.35±1.67 | 57.13±0.07 |
| 42793 | 75 | 4 | 40.00 | 71.25±3.07 | 23.12±3.22 | 34.38±5.44 |
| 43859 | 150 | 3 | 33.33 | 84.33±2.99 | 30.67±2.47 | 32.33±7.25 |
| 43875 | 150 | 3 | 33.33 | 85.33±2.34 | 32.00±2.34 | 42.33±7.34 |

## C.2 Ablation Analysis on OpenML Datasets

This section provides an ablation study based on the results of OpenML datasets in Table 1 and Figure 4. Both Table 1 and Figure 4 compares the performance of AI-IGL, CB, and IGL (full CI) on all OpenML datasets with balanced action distributions, and that with large sample size (sample size $\geq$ MNIST). We can observe that AI-IGL almost always outperforms IGL (full CI). We now demonstrate that the performance of AI-IGL is affected by: 1) the label noise; 2) the size of datasets (expected from theory), by the following observation.

At first, it is easy to notice that the performance of AI-IGL is significantly benefited from the sample size, by comparing with the performance of AI-IGL between all datasets and datasets with a large sample size (also suggested in Table 1), which follows the prediction of Theorem 2. On the other hand, we study how the label noise affects the performance of AI-IGL. To ablate the effect from the sample size, we only study the results in the dataset with large sample size. In this case, we

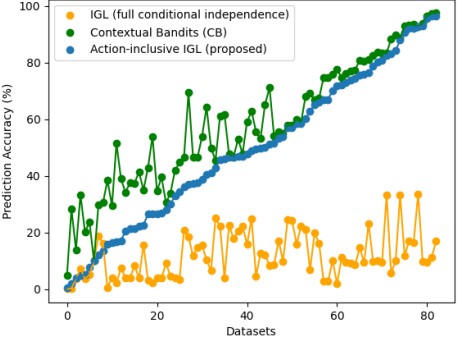
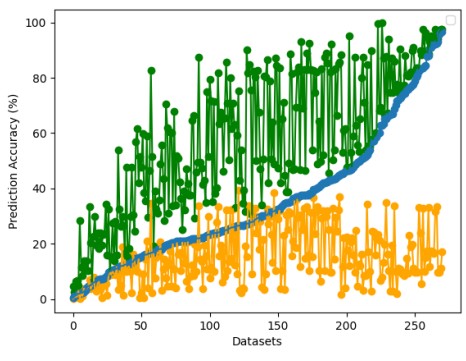

(a) Datasets with $K \geq 3$, $N \geq 70000$        (b) Datasets with $K \geq 3$, no constraints on $N$

Figure 4: Average performance on datasets with balanced action distributions from the OpenML benchmarking suite. For legibility of the figure, we do not include standard errors here. Standard errors for individual datasets are reported in Table 4. $K$ is the size of the action set and $N$ is the sample size.

consider the performance of CB to indicate the label noise, i.e., datasets with high CB performance means small label noise, and vice versa. We can observe from Figure 4(a) that the performance gap between CB and AI-IGL increases considerably with the label noise increasing (i.e., CB performance decreasing). This suggests that AI-IGL is more sensitive to the label noise compared with CB. It is unclear if this is an information-theoretical difficulty due to the setting without explicit reward, and we leave further investigation along this line as future work.

## C.3 Analyzing OpenML Dataset Meta-Properties with respect to AI-IGL's Performance

In order to better understand which features of a dataset make them amenable to high IGL performance, we systematically analyzed the features of OpenML datasets. Specifically, we measured a total of 15 features/meta-properties for all OpenML datasets and compared the accuracy of AI-IGL's success relative to CB. We first explain the first 12, and explanation for the additional 3 measures follow.

We collected a total of 49 features following the findings of Lorena et al. [2019]; Torra et al. [2008]; Reif et al. [2014]; Abdelmessih et al. [2010]. Based on preliminary analysis. We narrowed these features down to the following 12 features for further investigation of AI-IGL on OpenML datasets, because they had the least correlations with one another:

- Accuracy of the 1-nearest neighbor classifier on the dataset (1nn_accuracy) [Reif et al., 2014; Abdelmessih et al., 2010]

- Best single decision node accuracy created using the feature attribute with the highest information gain (best_node_accuracy) [Reif et al., 2014; Abdelmessih et al., 2010]

- The number of 0/1 features when the dataset is one-hot encoded (feature_onehot_count) [Lorena et al., 2019]

- Ratio of sample dimensionality by sample count (instance_per_feature) [Lorena et al., 2019]

- The ratio of the class distribution entropy and the maximum entropy for the uniform distribution over classes (class_entropy_N)

- Maximum Fisher discriminant ratio (max_fisher_discrim) [Lorena et al., 2019]

- The percentage of values in the feature matrix that are non-zero (max_single_feature_eff) [Lorena et al., 2019]

- Mutual information mean (mutual_XY_info_mean) [Torra et al., 2008; Reif et al., 2014]

- Naive Bayes accuracy (naive_bayes_accuracy) [Reif et al., 2014; Abdelmessih et al., 2010]

- Noise to signal ratio (noise_signal_ratio) [Torra et al., 2008]

- Number of principal components needed to represent 95% of data variability (pca_dims_95) [Lorena et al., 2019]
- Percent of data variance explained by the top principal component (pca_top_1_percent) [Lorena et al., 2019]

In addition to these 12 features, we added three additional features as follows. Compared to the typical CB guarantee, AI-IGL needs one more $K$ factor in its theoretical guarantees (Theorem 2). Since $K$ factors can be improved under some specific choice of function class (see discussion in Section 4), four additional features of the dataset were used to predict the relative performance of AI-IGL:

- $N$ (n)
- $N/\sqrt{K}$ (n_by_sqrt_k)
- $N/K$ (n_by_k)

We used a binary random forest classifier to predict the success of AI-IGL's performance relative to CB. If the relative performance is $\geq 0.7$, we label it as a success. In Table 5, we report the F1 scores for the success and failure classes defined in this way for a binary random forest classifier with 100 trees (evaluated using 10-fold cross validation on all 271 OpenML datasets). Based on further analysis of the importance weight for each feature computed using the information gain metric (Figure 5(a)), we created smaller subsets of the datasets to further analyze datasets representative of realistic interaction datasets with small sample sizes. We also report the average F1 scores for random forest classifiers trained and evaluated in a similar manner in rows 2,3 in Table 5.

| Dataset properties | AI-IGL accuracy$< 70\%$ | AI-IGL accuracy$\geq 70\%$ (Success) | Average F1 score (both classes) |
|---|---|---|---|
| $K \geq 3$ | 0.91 | 0.83 | 0.87 |
| $N/K \leq 1000$ | 0.95 | 0.46 | 0.71 |
| $N/K \leq 200$ | 0.97 | 0.56 | 0.77 |

Table 5: F1 scores of binary random forest classifiers predicting the success of AI-IGL relative to CB on different datasets using curated features/meta-properties.

We also computed the feature importance for each of the random forest classifiers to determine which feature is most predictive of AI-IGL's success relative to CB. Feature importance was computed using the information gain metric. We used python's scikit-learn library to implement the random forest classifiers as well as feature importance values.

We present feature importance plots for three different subsets of datasets in Figure 5. We find $N/K$ to be the most predictive feature of AI-IGL's relative performance (Figure 5(a)). It can alone predict its performance with an average F1 score of 0.79 under the same experimental setup. However, for datasets with a small value of $N/K$ ($\leq 1000, \leq 200$), there is high variability in relative performance. Using such a subset of datasets, we find maximum Fisher discriminant [Lorena et al., 2019] (a measure of classification complexity that quantifies the informativeness of a given sample) to be the most predictive of relative performance (Figure 5(b), Figure 5(c)).

This finding identifies clear measures of small and large datasets that can predict whether AI-IGL can match CB performance for any given dataset. It can also help researchers improve the design of novel applications of IGL, e.g., in HCI and BCI, by ensuring the resulting dataset's features are amenable to high performance.

For the features considered relatively more important than others as shown in Figure 5(a), 5(b) and 5(c)), we also present how the relative performance of AI-IGL varies for different values of such features. Visualizations for different features across all 271 datasets are shown in Figure 6, across the subset of 155 datasets with $N/K \leq 1000$ are shown in Figure 7, and across the subset of 128 datasets with $N/K \leq 200$ are shown in Figure 8.

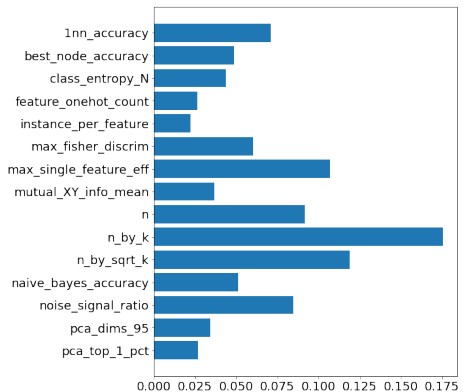

(a) Feature importance for 271 datasets with $K \geq 3$)

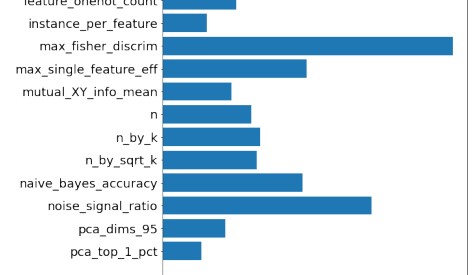

(b) Feature importance for 155 datasets with $N/K \leq 1000$

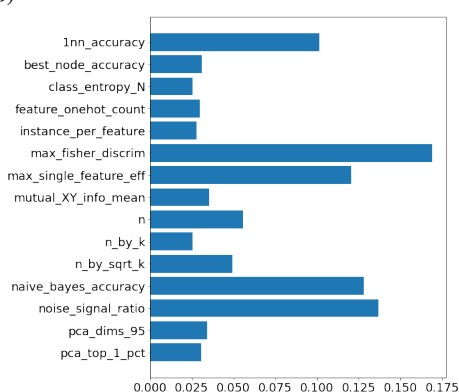

(c) Feature importance for 128 datasets with $N/K \leq 200$

Figure 5: Feature importance over dataset meta-features used to classify the success of AI-IGL. We find that for across all datasets $N/K$ is the most informative feature (Figure 5(a)), whereas maximum Fisher discriminant ratio is the most important feature over datasets with smaller values of $N/K$ (Figure 5(b) and 5(c)).

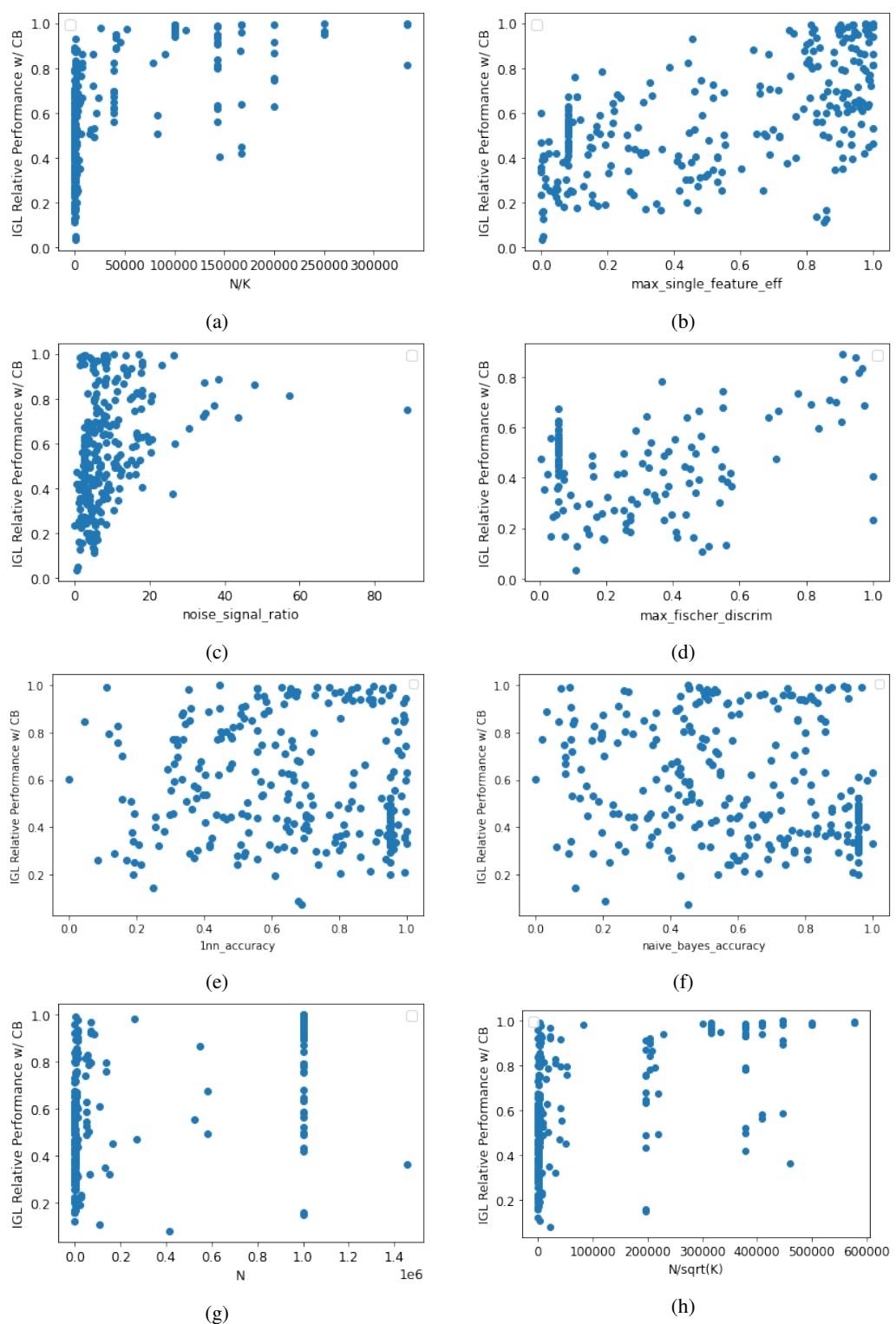

Figure 6: AI-IGL's relative performance w.r.t. CB versus different feature values for all 271 datasets with $K \geq 3$. $N/K$ is the most important feature for predicting AI-IGL's relative performance as shown in Figure 5(a). Here we observe that the variability in AI-IGL's relative performance decreases as $N/K$ increases. Datasets with a smaller values of $N/K$ are analyzed separately (discussed in Figure 7 and Figure 8).

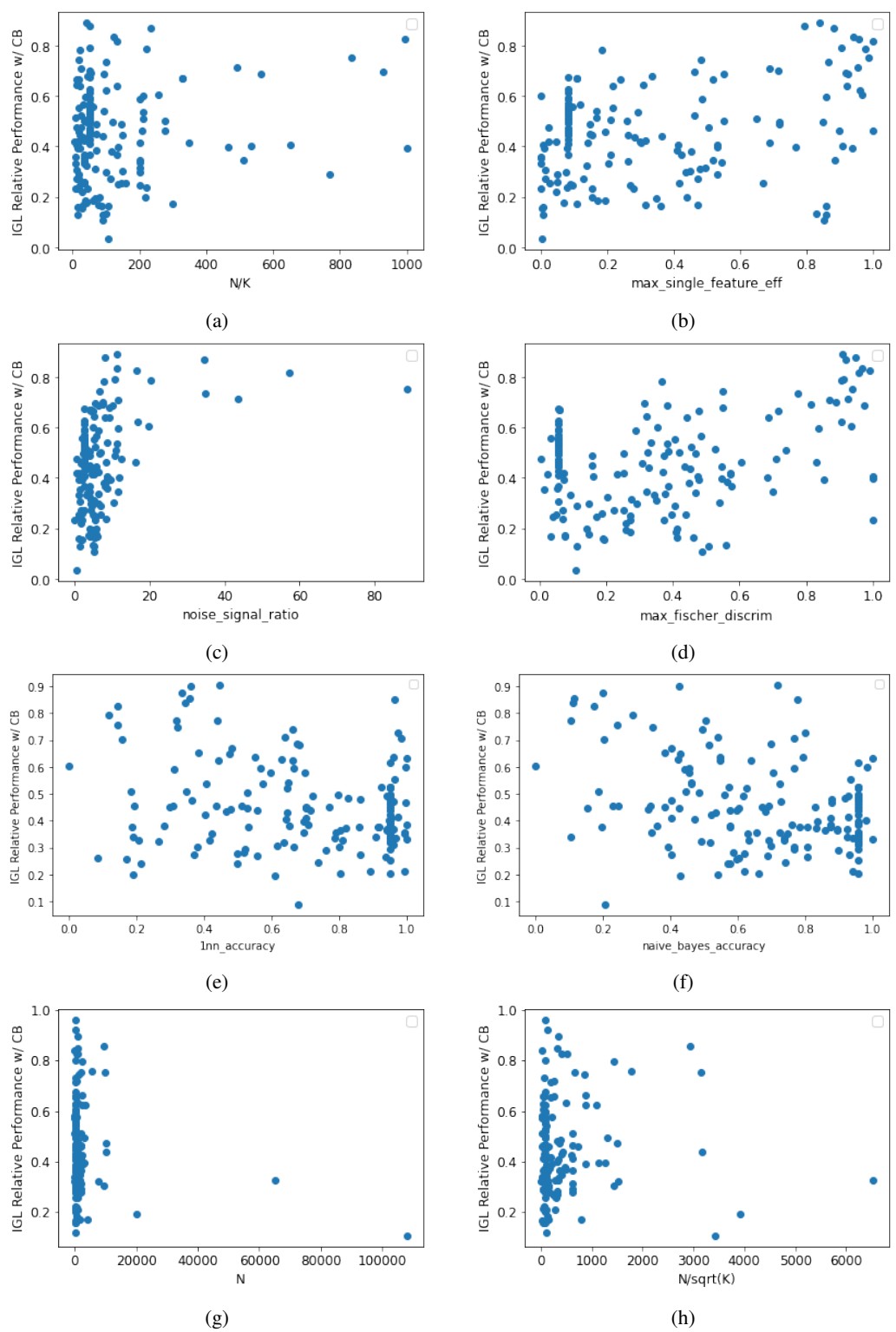

Figure 7: AI-IGL's relative performance w.r.t. CB versus different feature values for 155 datasets with $K \geq 3$ and $N/K \leq 1000$. $N/K$ itself is not the most important feature (Fig. 5(b)) for this subset of datasets, but the maximum Fisher discriminant ratio is which shows an approximately linear trend (Figure 7(d)).

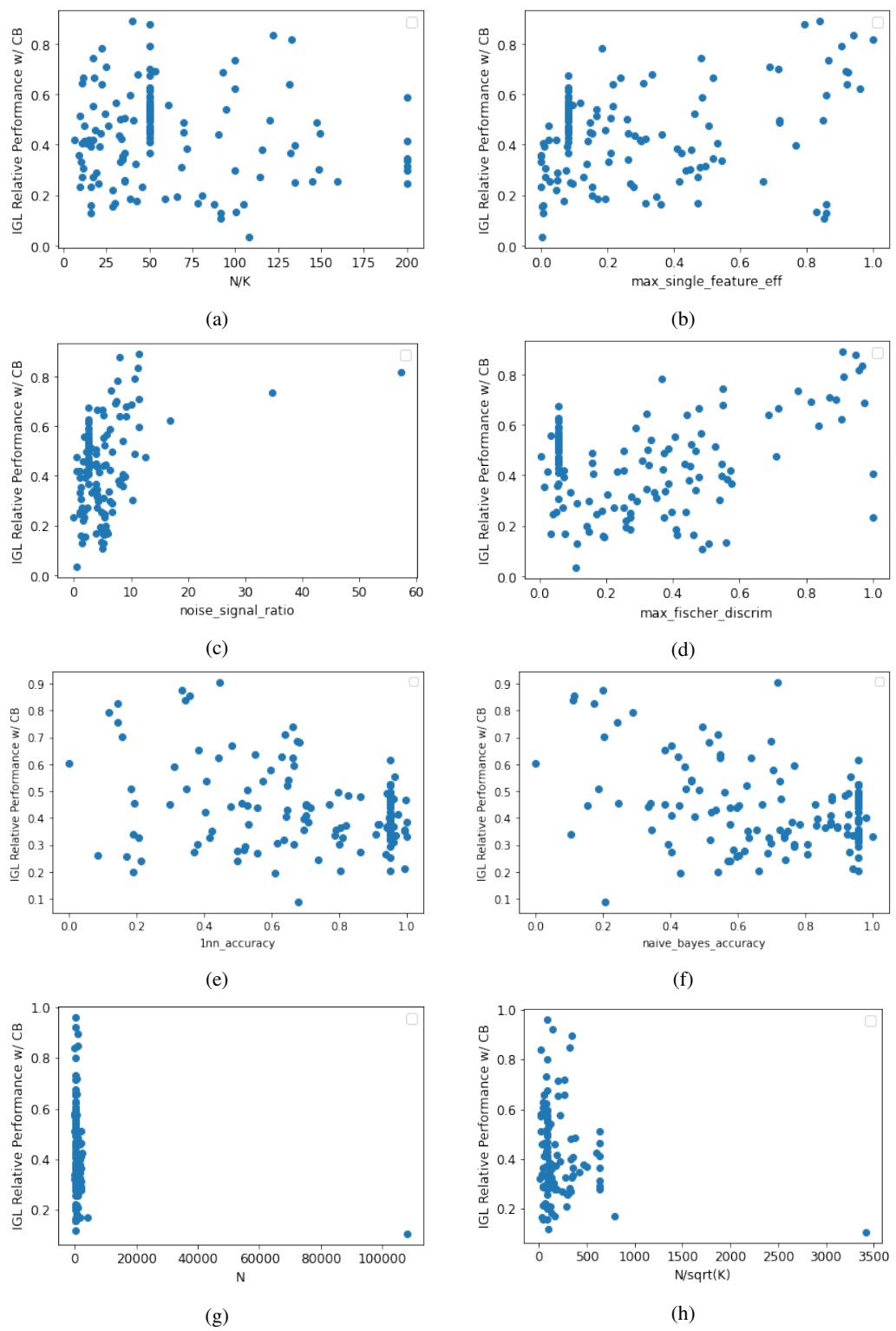

Figure 8: AI-IGL's relative performance w.r.t. CB versus different feature values for 128 datasets with $K \geq 3$ and $N/K \leq 200$. $N/K$ itself has a small importance value (Figure 5(c)) for this subset of datasets. The maximum Fisher discriminant ratio is again the most important feature for this subset of datasets (showing an approximately linear trend Figure 8(d)) to predict the relative performance of AI-IGL.