# OpenReview forum: "Interaction-Grounded Learning with Action-Inclusive Feedback"
_NeurIPS.cc/2022/Conference — NeurIPS 2022 Accept_

### Official Review · Reviewer_v1ct · 2022-07-10

**Rating:** 4
**Confidence:** 3
**Soundness:** 2 fair
**Presentation:** 3 good
**Contribution:** 1 poor

**Summary:**

This paper explores a straightforward reformulation of a newly introduced formalism — Interaction-Grounded Learning (IGL) — in which an agent must learn to act optimally in an environment only given access to some context, an action space, and per-step feedback that is some function of a latent reward function.

Whereas prior work formalizes the feedback component of IGL as conditionally independent of the context and language given the hidden rewards, this work (1) shows that this prior conditional independence assumption is too strong, leading to pathological failure modes when “action” information is present in the feedback, (2) presents a new formulation with less restrictive conditional independence assumptions that condition feedback on both latent reward AND executed action, and (3) presents a learning algorithm for deriving good policies given the new formulation and feedback.

Experiments on a modified MNIST task where the (context, action) are (image, label) respectively, and high-dimensional feedback is given as some image y where the digit shown is taken as (action + 6 * binary-reward - 3 modulo 10), and a similar construction for the various OpenML CC-18 datasets shows that learning with the proposed algorithm under the new conditional independence assumptions outperforms learning with the prior assumptions in cases where action-information is present in the feedback.

**Questions:**

- Notation in (1) is a bit unclear; L(\pi, \psi) seems to indicate a loss, but you’re actually trying to maximize the difference in Values right? Might it be better to call it J(\pi, \psi)?
- On line 115, it’s not clear where the augmented data is actually coming from; how are you generating the marginal distributions \mu(x) and\mu(y)? Especially if the data coming in is online interaction data... this feels rather strong? Or am I misunderstanding?
- In general - concrete examples to ground out the front half of the paper would be greatly appreciated!


**Limitations:**

In general this paper is severely limited by its evaluation. The paper seeks to show that the initial assumptions in the interaction-grounded learning framework are prohibitive and lead to failure modes, designing a new formulation of the framework’s assumptions, as well as a new learning algorithm to fix this. However, by only evaluating synthetic, toy constructed tasks with arbitrary feedback that is not realistic, it’s not clear that this is an actual problem.

This paper would be considerably stronger if evaluated on the motivating tasks in brain-computer interfaces or human-computer interaction used in the introduction and conclusion, and evaluating real-world instances of where action-intermingled-feedback is an actual problem.

**Strengths And Weaknesses:**

The key strength of this paper is in its clarity; at its core it takes an existing formalism (“full conditional independence Interaction-Grounded Learning”), finds a key failure mode, and adjusts the formalism with new, less restrictive assumptions. Each of the steps taken along the way are well-motivated, and the proofs tied to the new assumptions (separability, access to a baseline policy for symmetry breaking) are clean and relevant.

Unfortunately, the weaknesses of this paper are in the evaluation, and the underlying motivation behind this work; the introduction and discussion/conclusion motivate that this type of “Interaction Grounded Learning” is **critical** for scenarios in multimodal interactive feedback in human-computer interaction and in designing brain-computer interaction interfaces. Especially for the latter, the problem of having feedback data that is tied to action information is already a key problem in fMRI information orthogonalization (line 340), and for eye tracker recalibration for ALS patients (line 341). Unfortunately, none of the existing evaluations reflect these real-world use cases, instead constructing synthetic tasks based on MNIST or open classification benchmarks — tasks that are hard to understand.

For example, it’s hard to make the leap to these well-motivated, clear tasks from the current MNIST classification experiments where feedback is an arbitrary formula of the “right label” and the “predicted action” — in the paper experiments, the feedback provided an agent is a digit with label = (action + 6 * binary-reward - 3 modulo 10). In general, it’s not clear why the action-intermingling with feedback is an action problem in real-world settings, or why other, simpler approaches for learning couldn’t learn to decouple this information.

EDIT: The new revision does address this a little bit, so I am updating my score.

---

> ### Author Response · Authors · 2022-08-02
> **Author Response to Reviewer v1ct**
>
> We would like to thank the reviewer for their constructive comments on the manuscript. We especially thank the reviewer for their positive feedback on our clarity in writing. We would like to reiterate that our paper focuses on the *theory and methodology* of IGL with action-inclusive feedback, which is motivated by real-world applications.
>
> We address the concerns raised by the reviewer below. Please let us know if further clarification is needed.
>
> ------
>
> > Motivation for AI-IGL and why action-intermingling with feedback is a concern in realistic scenarios ["In general, it's not clear why the action-intermingling with feedback is an action problem in real-world settings"]
>
> Thank you for pointing out the writing suggestions, we are sure they will increase the readability of the paper. We have updated the introduction by providing motivation for realistic BCI and HCI tasks (changes highlighted in red in the updated draft). We also provide additional results for a number-guessing BCI experiment (please check Sec. 5.2).
>
> ------
>
> > "why other, simpler approaches for learning couldn’t learn to decouple this information"
>
> To our best knowledge, there is no other approach that is provably able to decouple the reward information from action-inclusive feedback in the IGL setting. The close-related work [Xie et al. 2021] is shown to fail both theoretically (Example 3) and empirically in our paper.
>
> ------
>
> > "Notation in (1) is a bit unclear"
>
> We thank the reviewer for their suggestion to improve the notation for Equation 1. We modified this notation in red in our revision.
>
> ------
>
> > "On line 115, it’s not clear where the augmented data is actually coming from"
>
> In the online setting, one can maintain the historical data for y, and then draw an independent y for the current x. Also, the requirement of augmented data can be dropped via spectral contrastive loss discussed in Section 3.1.
>
> ------
>
> > Grounding with examples for the first half of the paper
>
> We have updated the introduction of the draft by grounding our motivation to realistic experiment settings (see changes in red). We also follow up with results from one such BCI experiment using a 3rd party fMRI simulator [Ellis et al. 2020, Sec 5.2].
>
> ------
>
> > "evaluating synthetic, toy constructed tasks"
>
> There may be a misunderstanding here. The 271 datasets we used are real datasets from OpenML.  The exact feedback mechanism is synthetic, but that is unavoidable if you value a reproducible demonstration of the robustness of the approach across many different datasets.
>
> ------
>
> > "It's not clear that this is an actual problem"
>
> Given the causal order of observation, action, and reaction, tasks with action information intermingled into the reaction are pervasive from basic principles.
>
> ------
>
> > "This paper would be considerably stronger if evaluated on the motivating tasks in brain-computer interfaces or human-computer interaction..."
>
> We disagree with the premise here.  It is normal at NeurIPS to engage in multi-year multi-paper research projects rather than requiring an entire research project be contained within a single paper.  This norm goes back to Terry Sejnowski (the president of NeurIPS) who commonly says “one paper, one idea”.  This norm is extremely valuable, because it allows the community to be aware of, comment on, and reuse partially completed research programs.  We would encourage you to consider and evaluate according to this community norm.  Evaluating according to this norm: “Is enabling action inclusive feedback a significant step towards enabling BCI & HCI applications?”  We believe a fair answer is ‘yes’, given that it is obviously essential from basic principles.
>
> ------
>
> **Reference:**
>
> Tengyang Xie, John Langford, Paul Mineiro, and Ida Momennejad. Interaction-grounded learning. In International Conference on Machine Learning, pages 11414–11423. PMLR, 2021.
>
> Cameron T Ellis, Christopher Baldassano, Anna C Schapiro, Ming Bo Cai, and Jonathan D Cohen. Facilitating open-science with realistic fmri simulation: validation and application. PeerJ, 8:e8564, 2020.

---

> > ### Comment · Reviewer_v1ct · 2022-08-08
> > **Rebuttal Response**
> >
> > I'd like to thank the authors for their rebuttal! I'm sorry for the late response, after the rebuttal it became clear that I needed to familiarize myself with prior work in the field, because the authors are correct - my understanding and the way I evaluate this work may not necessarily be grounded in current practice.
> >
> > I think the bulk of my initial negative review can be seen as a mismatch between the fundamental experiments presented in this work, vs. a larger, more applied study that seemed to be hinted at in the earlier parts of the paper (e.g., looking at BCI/HCI applications more explicitly).
> >
> > I really appreciate the time the authors took to add a more grounded discussion and extra experiments to the paper, but I do want to address the final point in the rebuttal above, which I don't really agree with:
> >
> > > It is normal at NeurIPS to engage in multi-year multi-paper research projects rather than requiring an entire research project be contained within a single paper. This norm goes back to Terry Sejnowski (the president of NeurIPS) who commonly says “one paper, one idea”. This norm is extremely valuable, because it allows the community to be aware of, comment on, and reuse partially completed research programs.
> >
> > I strongly *do* believe in this point - and at the same time, don't want to fall into a false dichotomy here; if the goal of this paper is to demonstrate that IGL solves a real-world problem, this paper does need to 1) show that this is a problem in the real-world (e.g., by examining real-world data), 2) present an approach that could solve the problem (which is done via the current evaluation), and 3) actually show that the proposed approach can generalize to the real-world problems introduced in (1). I think the new experiments do help address (1) and (3) a little bit, and I am updating my score accordingly, however, I would like this work to shift beyond the toy MNIST experiment that remains at the focus of the evaluation.

---

> > > ### Comment · Reviewer_ejKG · 2022-08-09
> > > **Comment**
> > >
> > > I am unfortunately also late to this, but a question for the authors is what kind of effort would be required to actually perform human-agent or BCI studies, i.e. in terms of cost and time? My understanding (having done a few human studies myself) is that these kind of experiments are extremely expensive, time-consuming, and stochastic, not to mention in the case of BCI, potentially very invasive (disclaimer: I don't know much about BCI research).
> > >
> > > I think the premise of the work is that this is a general problem arising in BCI / HCI and other sequential decision processes with the same properties as formally defined in the paper. That general problem can be studied outside of a human experiment using synthetic data, as a preliminary approach. I do think this is a limitation of the paper, and generally I also care a lot about real-world evaluation of proposed systems, but if this is a general problem that arises in different sequential decision processes, maybe a first step is showing more theoretical results in a controlled experiment. Another question for authors: is there some kind of dataset derived from a BCI/HCI study you could use to simulate real-world performance? (I would guess this is not useful if so, because the data you train on is so policy-dependent)

---

> > > > ### Author Response · Authors · 2022-08-09
> > > > **Author Response**
> > > >
> > > > We thank the reviewer for correctly pointing out that the high cost and effort of BCI experiments [1] and what it entails: that it is worthwhile to make theoretical progress tested on synthetic data. We further agree that general problems can be studied with significant advances using synthetic data, and the closer the synthetic data fits with the real-world scenarios, the ultimate human experiment will be more optimally designed.
> > > >
> > > > As to the reviewer's question regarding datasets, the reviewer asks, "*is there some kind of dataset derived from a BCI/HCI study you could use to simulate real-world performance?*". We have seriously considered this and identified candidate datasets. However, while we have plans for using existing real-world datasets, the reviewer is correct that the interactive nature of AI-IGL (and the IGL category) requires the reaction/feedback signals to specifically address the guess made by the system. Clearly, that requires our own human-designed experiment. While we are actively pursuing this direction, we fully agree with the reviewer that publishing theoretical advances has its own merits, especially with closely simulated synthetic data as our novel experiment. We believe that our realistic fMRI simulation of an IGL experiment inspired by real fMRI data accomplishes this (Sec 5.2 and Fig. 1), and can serve as an example of designing realistic BCI experiments with realistic neural simulations to compare theoretical proposals and algorithmic advances. The design and analysis of novel human BCI experiments merit a separate publication, especially given the convention of "one idea per paper" at NeurIPS.
> > > >
> > > > **Reference:**
> > > >
> > > > [1] Debettencourt, Megan T., Jonathan D. Cohen, Ray F. Lee, Kenneth A. Norman, and Nicholas B. Turk-Browne. "Closed-loop training of attention with real-time brain imaging." Nature Neuroscience 18, no. 3 (2015): 470-475.

---

### Official Review · Reviewer_ejKG · 2022-07-11

**Rating:** 7
**Confidence:** 1
**Soundness:** 4 excellent
**Presentation:** 4 excellent
**Contribution:** 4 excellent

**Summary:**

This paper studies the problem of interaction-grounded learning, where an agent takes an action a given context x, and receives a feedback vector y in return. A reward (to be optimized) is also generated from the environment, but not revealed to the learning process. The goal is to learn a policy that optimizes reward without explicitly observing reward from the environment, by learning a mapping from Y to the space of rewards, and learning a value function mapping X and A to [0, 1]. An existing challenge of this setting is that IGL fails when y contains information about the executed action a. This paper proposes a contrastive learning approach that allows IGL to succeed even when y contains information about a. Experiments on several RL benchmarks show improvements over IGL assuming conditional independence of y given a, nearing performance of contextual bandit (which assumes full access to reward).

**Questions:**

Questions about some other assumptions that seem to be made in the paper:
* Does this algorithm / setting assume a discrete action space? What happens if the action space is continuous?
* Does the setting assume that y is deterministic given (x, a)? It seems like no, given the experiments.
* Is there an assumption that there exists an optimal \Psi, that can map any (y, a) to the correct reward? I suppose if y is deterministic given r, then yes. But it seems like that's not the case given the experiments.
* Does the algorithm require executing all actions in all states in the batch data D? If so, seems like another assumption that might break in real-world settings.
* In Table 1, action-exclusive feedback is just the image of the reward, as in Figure 2?
* Images y in 5.1 are randomly chosen from the dataset?

**Limitations:**

See above about limitations / assumptions for applications to real-world settings.

**Strengths And Weaknesses:**

Strengths:

* Paper is relatively clear and easy to follow. As someone who doesn't have much practice with reading theory-heavy papers (unfortunately this paper is a few steps out of my area), I found it easy to follow. I found the information-theoretic arguments intuitive.
* Main contributions have potential broad significance (e.g., for applications as discussed in the introduction)

---

Weaknesses:

* See assumptions below -- discussion on unstated assumptions (if any) and their limits on real-world applications would be nice to have.
* I found the format of Figure 2 somewhat confusing; to me the y-axis of the graphs seems to imply some kind of sequential process

---

> ### Author Response · Authors · 2022-08-02
> **Author Response to Reviewer ejKG**
>
> We would like to thank the reviewer for their constructive comments on the manuscript. We especially thank the reviewer for their positive feedback on our clarity in writing, intuitive information-theoretic arguments, explanation of the novel paradigm of IGL, as well how our main technical contribution AI-IGL which solves IGL under more complex feedback settings in comparison to prior work in this space.
>
> We address the concerns raised by the reviewer below. Please let us know if further clarification is needed.
>
> ------
>
> > "Does the setting assume that $y$ is deterministic given $(x, a)$?"
>
> No, $y$ is not deterministic given $(x,a)$. For example, in our MNIST experiment, $(x,a)$ corresponds to a deterministic latent reward r, but the emission process from the latent reward (digit of r) to its corresponding image is stochastic. Note that the algorithm provably works in the case where both latent reward and emission are stochastic.
>
> ------
>
> > Optimality of Psi ["Is there an assumption that there exists an optimal $\Psi$, that can map any $(y, a)$ to the correct reward?"]
>
> Yes, this is assumed by the separability assumption (Assumption 3).
>
> ------
>
> > "Does the algorithm require executing all actions in all states in the batch data $\mathcal D$?"
>
> In all of our experiments, we only execute one action for each observation.  Of course, each latent state is typically represented in multiple observations.
>
> ------
>
> > Action-exclusive feedback in Table 1 ("In Table 1, action-exclusive feedback is just the image of the reward, as in Figure 2?")
>
> Yes, the reviewer’s understanding is correct. Action-exclusive feedback refers to the case of full CI IGL [Xie et al, 2021] where the feedback is the image of the binary reward ($0$ or $1$), and no action information is present in the feedback. Please note that the original Figure 2 is now Figure 3 in the Appendix.
>
> ------
>
> > Images for feedback in the MNIST experiment ("Images $y$ in 5.1 are randomly chosen from the dataset?")
>
> The images for the feedback are chosen based on the formula $y = (a+6r-3) \mod 10$. It is exactly similar to how the feedback images are generated for Figure 3(c) (in the updated draft).
>
> ------
>
> > Confusion about previous Figure 2 ("Does the $y$-axis imply some kind of sequential process?")
>
> The previous Figure 2 for the MNIST example is now Figure 3 in the Appendix. The reviewer’s understanding is correct. Interactive learning with IGL is a sequential learning process. Primarily, the figure is meant to indicate the format of the feedback information available to learning agents of different algorithms and how they compare to one another.
>
> ------
>
> > Limits of assumptions on real-world applications
>
> Our proposed solution relaxes a stringent assumption on prior work (full CI IGL), moving us closer to realistic applications.

---

> > ### Comment · Reviewer_ejKG · 2022-08-09
> > **Thank you**
> >
> > Thank you for the response.

---

### Official Review · Reviewer_R7aA · 2022-07-12

**Rating:** 5
**Confidence:** 2
**Soundness:** 3 good
**Presentation:** 2 fair
**Contribution:** 2 fair

**Summary:**

The paper looks at the setting where the goal is to learn a policy without access to a reward from the environment, with access only to feedback. Previous work (Xie et al (2021)) assumes that the feedback does not have the action. This paper removes that assumption and proposes a new contrastive approach that can integrate feedback with actions to learn a policy. Through different empirical evaluations (toy MNIST task, and large-scale synthetic eval on OpenML), the authors show how this approach can learn from feedback that has the actions taken by the agent.

**Questions:**

How does this work relate to learning from interaction works in robotics / HRI and human-in-the-loop learning?

**Limitations:**

The authors’ discuss the potential negative impact of the work well. Limitations of the approach and testing might be lacking; see weaknesses.

**Strengths And Weaknesses:**

Strengths

- The paper removes some assumptions introduced in previous works to bring algorithms closer to being realised in realistic environments.
- Learning from non-scalar rewards and feedback is an important direction and this work makes progress towards making this possible.
- The paper has the appropriate ablations and model comparisons.

Weaknesses

I am not very familiar with the relevant literature in the subfield. As someone from a related but outside the subfield, here are some potential weaknesses. My main concerns lie with the writing of the paper.

- Paper is not clearly written. The introduction does not motivate the central problem well. I would have liked to see a few examples where previous methods fail, but the paper is able to bridge the gap. The authors add these to the conclusion, but it might be useful to add examples to the introduction to motivate the problem. Section 3 is especially difficult to understand as there is a lot of notation with little intuition.
- The empirical evaluations aren’t described well. I had to refer to Xie et al. (2021) to get a better idea of what the evaluations look like. Similarly, the previous paper also does a much better job of describing and motivating different applications.
- The empirical evaluations are limited; assuming very simple policies and representations. These would be difficult to realize in realistic settings that the authors use as examples.
- The intuition behind the feedback signal for the MNIST classification signal is not clear.
- The empirical evaluations seem detached from potential practical applications.
- Feedback is often noisy; the authors do not consider this scenario.
- There is a lack of discussion of why a CB setting is better than an MDP formulation for the tasks proposed.

---

> ### Author Response · Authors · 2022-08-02
> **Author Response to Reviewer R7aA**
>
> We would like to thank the reviewer for their constructive comments on the manuscript. We address the concerns raised by the reviewer below. Please let us know if further clarification is needed.
>
> ------
>
> > Writing (Motivation) ["The introduction does not motivate the central problem well."]
>
> Thank you for pointing out the writing suggestions, we are sure they will increase the readability of the paper. We have made the suggested writing corrections in the paper. We updated the introduction by providing motivation for several realistic applications (changes highlighted in red in the updated draft). We ground the motivation with a number-guessing BCI experiment, for which we also provide additional results in the manuscript (Please check Sec. 5.2).
>
> ------
>
> > Writing ["The empirical evaluations are not described well"]
>
> We thank the reviewer for taking the time to read prior literature relevant to our work. Similar to the motivating example from Xie et al. [2021], we include a new figure (Figure 1) in the paper for a BCI experiment where the feedback is a human’s fMRI signal (coupled action and latent reward information). We also include results for such a simulated task for which data is generated using a publicly available realistic 3rd party simulator (Ellis et al. 2020) (please refer to Sec. 5.2  in the updated draft of the paper).
>
> ------
>
> > Realistic Experiments ["The empirical evaluations are limited; assuming very simple policies and representations. These would be difficult to realize in realistic settings that the authors use as examples."]
>
> Large scale experiments with a synthetic feedback signal provide a reproducible demonstration that it is possible to robustly ground learning as the AI-IGL theory suggests.   We agree that more specific experiments with a more realistic dataset are valuable as well, so with the new results both general robustness and specific applicability are demonstrated.
>
> ------
>
> > "The intuition behind the feedback signal for the MNIST classification signal is not clear"
>
> We have moved the MNIST experiment to Appendix B in the updated draft of the paper. We expand on the motivation for the MNIST feedback signal in the introduction as well as with an additional BCI experiment for a number-guessing task in Sec 5.2 (please also see Figure 1).
>
> ------
>
> > "Feedback is often noisy; the authors do not consider this scenario"
>
> We have added results in Sec 5.2 which show that AI-IGL also succeeds compared to full CI IGL under noisy feedback.
>
> ------
>
> > "There is a lack of discussion of why a CB setting is better than an MDP formulation for the tasks proposed")
>
> Please note we are not claiming that comparison to CB is better than comparison to an MDP setting. The MDP setting is more challenging than CB (due to the credit assignment problem). However, the original IGL framework [Xie et al. 2021] is developed with respect to a CB approach, and our work builds on that formulation.
>
> ------
>
> > "How does this work relate to learning from interaction works in robotics / HRI and human-in-the-loop learning?"
>
> Prior work on interactive reinforcement learning [Li et al. 2019] assumes that during interaction, the human provides a scalar numeric reward to guide the agent’s learning. Interaction grounded learning is a more complex setting where the human is not using a clicker-style device to provide numeric feedback, but rather reacting naturally to the learning agent’s actions through implicit feedback signals. A recent work on learning from implicit human feedback [Cui et al. 2020] assumes a pretraining (supervised learning) phase where the implicit feedback is first grounded to the latent reward, and then a traditional reinforcement learning algorithm is used with the decoded reward from the first phase. On the other hand, interaction grounded learning is a natural interactive paradigm making no assumptions about a pre-training stage (such a phase can definitely accelate the learning of IGL but is not necessary to succeed eventually).
>
> ------
>
> **Reference:**
>
> Tengyang Xie, John Langford, Paul Mineiro, and Ida Momennejad. Interaction-grounded learning. In International Conference on Machine Learning, pages 11414–11423. PMLR, 2021.
>
> Cameron T Ellis, Christopher Baldassano, Anna C Schapiro, Ming Bo Cai, and Jonathan D Cohen. Facilitating open-science with realistic fmri simulation: validation and application. PeerJ, 8:e8564, 2020.
>
> Li G, Gomez R, Nakamura K, He B. Human-centered reinforcement learning: A survey. IEEE Transactions on Human-Machine Systems. 2019 May 7;49(4):337-49.
>
> Yuchen Cui, Qiping Zhang, Brad Knox, Alessandro Allievi, Peter Stone, and Scott Niekum. The empathic framework for task learning from implicit human feedback. In Conference on Robot Learning, pages 604–626. PMLR, 2021.

---

### Author Response · Authors · 2022-08-03
**Summary of Changes in the Revision**

We thank all reviewers for their insightful comments. We uploaded a revision of our paper, where we addressed the concerns raised by each reviewer carefully. We would like to summarize the changes in our revision here.

Summary of the revision:
- We provided a schematic BCI example of the problem studied in our paper (Figure 1).
- We added a new experiment based on a realistic fMRI simulator and real fMRI data from humans (Section 5.2), which simulates the schematic BCI example.
- We fixed the minor notations issues in red.

---

### Author Response · Authors · 2022-08-09
**Thank the Reviewers**

As the discussion period closes, we thank the reviewers for their attention. The paper has definitely improved due to the constructive feedback. We are especially pleased with the more realistic simulation, which was the direct result of reviewer comments.

---

### Meta-Review · Area_Chair_zHh2 · 2022-08-28

**Recommendation:** Accept
**Confidence:** Certain

**Metareview:**

This paper addresses the problem of learning to behave optimally when actions result only in new observations but no rewards. Feedback is provided in the shape of a vector. This problem, known as IGL, has already been described in previous works which had to make the assumption that the action was not included in the feedback. This paper gets rid of this assumption and provides theoretical guarantees.

The discussion has been quite extensive and the main issue raised by reviewers concerned the experimental setups. They were considered toy-ish and too far from a real application. Especially, the authors mentioned BCI and HCI in their intro (mainly focusing on the fact that having the action in the observation is mandatory with humans in the loop) but didn't provide experiments involving actual BCI or HCI.

The authors tried to address this issue by providing synthetic experiments simulating BCI and fMRI. As the authors stated, the cost of real experiments in that setup would be prohibitive.

Given the effort made by authors to provide experimental results supporting them, the algorithmic and theoretical contributions seem good enough to reach the acceptance bar.

**Award:**

No

---

### Decision · Program_Chairs · 2022-09-14

Accept